# RAPHAEL: Text-to-Image Generation via Large Mixture of Diffusion Paths

**Zeyue Xue**[*]
The University of Hong Kong
xuezeyue@connect.hku.hk

**Guanglu Song**[*]
SenseTime Research
songguanglu@sensetime.com

**Qiushan Guo**
The University of Hong Kong
qsguo@cs.hku.hk

**Boxiao Liu**
SenseTime Research
liuboxiao@sensetime.com

**Zhuofan Zong**
SenseTime Research
zongzhuofan@gmail.com

**Yu Liu**[†‡]
SenseTime Research
liuyuisanai@gmail.com

**Ping Luo**[‡]
The University of Hong Kong
Shanghai AI Laboratory
pluo@cs.hku.hk

*"When one is painting one does not think."*

*— Raffaello Sanzio da Urbino*

## Abstract

Text-to-image generation has recently witnessed remarkable achievements. We introduce a text-conditional image diffusion model, termed RAPHAEL, to generate highly artistic images, which accurately portray the text prompts, encompassing multiple nouns, adjectives, and verbs. This is achieved by stacking tens of mixture-of-experts (MoEs) layers, *i.e.,* space-MoE and time-MoE layers, enabling billions of diffusion paths (routes) from the network input to the output. Each path intuitively functions as a "painter" for depicting a particular textual concept onto a specified image region at a diffusion timestep. Comprehensive experiments reveal that RAPHAEL outperforms recent cutting-edge models, such as Stable Diffusion, ERNIE-ViLG 2.0, DeepFloyd, and DALL-E 2, in terms of both image quality and aesthetic appeal. Firstly, RAPHAEL exhibits superior performance in switching images across diverse styles, such as Japanese comics, realism, cyberpunk, and ink illustration. Secondly, a single model with three billion parameters, trained on $1,000$ A100 GPUs for two months, achieves a state-of-the-art zero-shot FID score of 6.61 on the COCO dataset. Furthermore, RAPHAEL significantly surpasses its counterparts in human evaluation on the ViLG-300 benchmark. We believe that RAPHAEL holds the potential to propel the frontiers of image generation research in both academia and industry, paving the way for future breakthroughs in this rapidly evolving field. More details can be found on a webpage: `https://raphael-painter.github.io/`[§].

---

[*]Equal contribution. Work done during Zeyue's internship at SenseTime Research.

[†]Project lead.

[‡]Corresponding authors.

[§]More creations can be found in `https://miaohua.sensetime.com/zh-CN/picture-selection`. Please select the Artist v0.3.5 model to generate. This is our latest version based on RAPHAEL. This information was last updated on Oct. 16th, 2023.

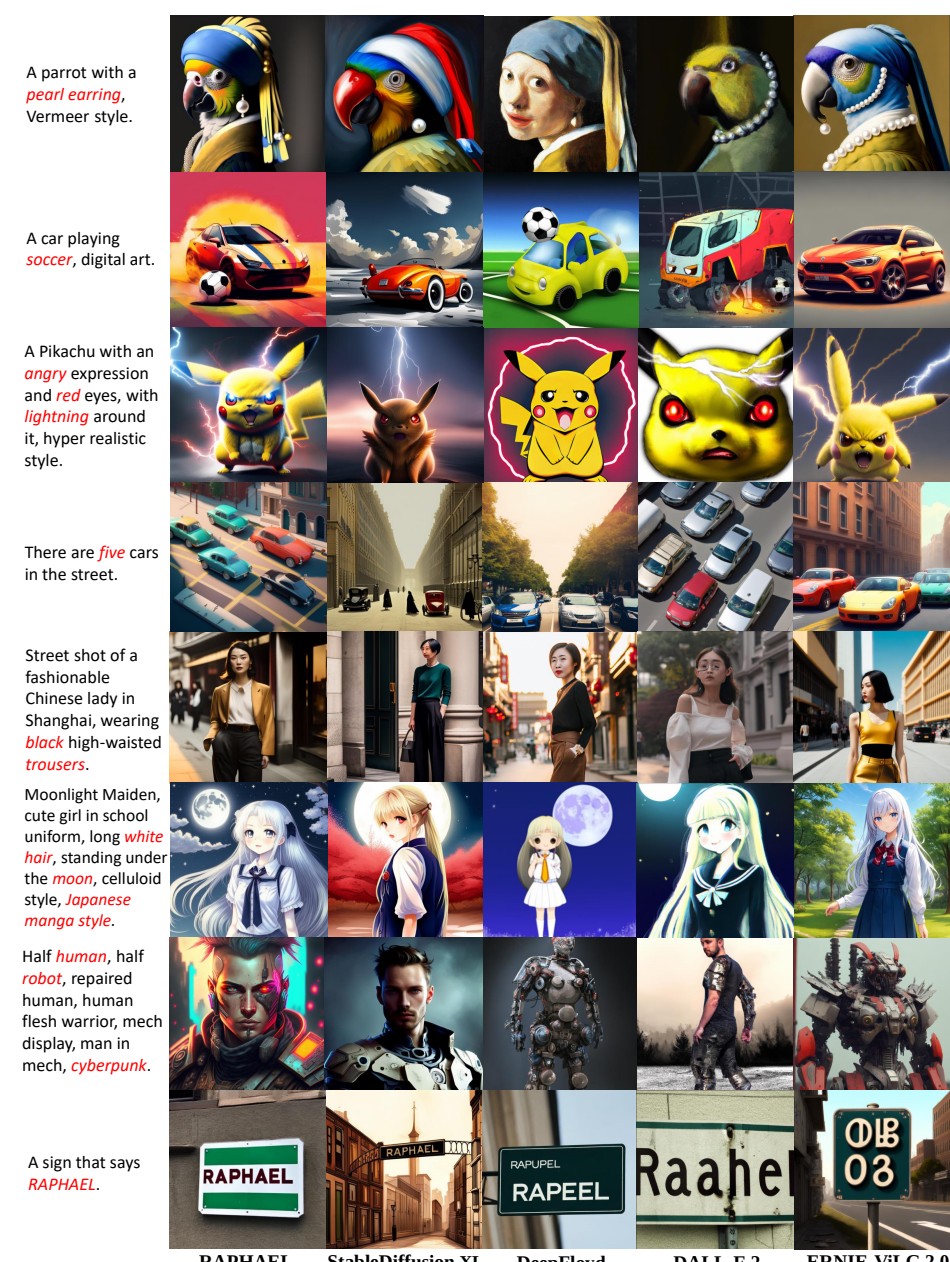

**A parrot with a *pearl earring*, Vermeer style.**

**A car playing *soccer*, digital art.**

**A Pikachu with an *angry* expression and *red* eyes, with *lightning* around it, hyper realistic style.**

**There are *five* cars in the street.**

**Street shot of a fashionable Chinese lady in Shanghai, wearing *black* high-waisted *trousers*.**

**Moonlight Maiden, cute girl in school uniform, long *white hair*, standing under the *moon*, celluloid style, *Japanese manga style*.**

**Half *human*, half *robot*, repaired human, human flesh warrior, mech display, man in mech, *cyberpunk*.**

**A sign that says *RAPHAEL*.**

| RAPHAEL | StableDiffusion XL | DeepFloyd | DALL-E 2 | ERNIE-ViLG 2.0 |

Figure 1: **Comparisons** of RAPHAEL with recent representative generators, Stable Diffusion XL [2], Deep-Floyd, DALL-E 2 [3], and ERNIE-ViLG 2.0 [5]. They are given the same prompts, where the words that the human artists yearn to preserve within the generated images are highlighted in red. These images are not cherry-picked. We see that previous models often fail to preserve the desired concepts. For example, only the RAPHAEL-generated images precisely reflect the prompts such as "pearl earring, Vermeer", "playing soccer", "five cars", "black high-waisted trouser", "white hair, manga, moon", and "sign, RAPHAEL", while other models generate compromised results. **Better zoom in 200%**.

## 1 Introduction

Recent advancements in text-to-image generators, such as Imagen [1], Stable Diffusion [2], DALL-E 2 [3], eDiff-I [4], and ERNIE-ViLG 2.0 [5], have yielded remarkable success and found wide applications in computer graphics, culture and art, and the generation of medical and biological data.

Despite the substantial progress made in text-to-image diffusion models [1, 2, 3, 4, 5], there remains a pressing need for research to further achieve more precise alignment between text and image. As

illustrated in Fig.1, existing models often fail to adequately preserve textual concepts within the generated images. This is primarily due to the reliance on a classic cross-attention mechanism for integrating text descriptions into visual representations, resulting in relatively coarse control of the diffusion process, and leading to compromised results.

To address this issue, we introduce RAPHAEL, a text-to-image generator, which yields images with superior artistry and fidelity compared to prior work, as demonstrated in Fig.2. RAPHAEL, an acronym that stands for "distinct image **r**egions **a**lign with different text **ph**ases in **a**ttention **l**earning", offers an appealing benefit not found in existing approaches.

Specifically, we observe that different text concepts influence distinct image regions during the generation process [6], and the conventional cross-attention layer often struggles to preserve these varying concepts adequately in an image. To mitigate this issue, we employ a diffusion model stacking tens of mixture-of-experts (MoE) layers [7, 8], including both space-MoE and time-MoE layers. Concretely, the space-MoE layers are responsible for depicting different concepts in specific image regions, while the time-MoE layers focus on painting these concepts at different diffusion timesteps.

This configuration leads to billions of diffusion paths from the network input to the output. Naturally, each path can act as a "painter" responsible for rendering a particular concept to an image region at a specific timestep. The result is a more precise alignment between text tokens and image regions, enabling the generated images that accurately represent the associated text prompt. This approach sets RAPHAEL apart from existing models and even sheds light on future studies of the explainability of the generation process. Additionally, we propose an edge-supervised learning module to further enhance the image quality and aesthetic appeal of the generated images.

Extensive experiments demonstrate that RAPHAEL outperforms preceding approaches, such as Stable Diffusion, ERNIE-ViLG 2.0, DeepFloyd, and DALL-E 2. (1) RAPHAEL exhibits superior performance in switching images across diverse styles, such as Japanese comics, realism, cyberpunk, and ink illustration. (2) RAPHAEL establishes a new state-of-the-art with a zero-shot FID-30k score of 6.61 on the COCO dataset. (3) RAPHAEL, a single model with three billion parameters trained on $1,000$ A100 GPUs, significantly surpasses its counterparts in human evaluation on the ViLG-300 benchmark.

The **contributions** of this work are three-fold: **(i)** We propose a novel text-to-image generator, RAPHAEL, which, through the implementation of several carefully-designed techniques, generates images that more accurately reflect textual prompts than previous works. **(ii)** We thoroughly explore RAPHAEL's potential for switching images in diverse styles, such as Japanese comics, realism, cyberpunk, and ink illustration, and for extension using LoRA [9], ControlNet [10], and SR-GAN [11]. **(iii)** We will release a programming API for RAPHAEL to the public. We believe that RAPHAEL holds the potential to advance the frontiers of image generation in both academia and industry, paving the way for future breakthroughs in this rapidly evolving field.

## 2  Notation and Preliminary

We present the necessary notations and the Denoising Diffusion Probabilistic Model (DDPM) [12] for text-to-image generation. Given a collection of $N$ images, denoted as $\{\mathbf{x}_i\}_{i=1}^N$, the aim is to learn a generative model, $p(\mathbf{x})$, that is capable of accurately representing the underlying distribution.

In forward diffusion, Gaussian noise is progressively introduced into the source images. At an arbitrary timestep $t$, it is possible to directly sample from the Gaussian distribution following the $T$-step noise schedule $\{\alpha_t\}_{t=1}^T$, without iterative forward sampling. Consequently, the noisy image at timestep $t$, denoted as $\mathbf{x}_t$, can be expressed as $\mathbf{x}_t = \sqrt{1 - \bar{\alpha}_t}\mathbf{x}_0 + \sqrt{\bar{\alpha}_t}\epsilon_t$, where $\bar{\alpha}_t = \prod_{i=1}^t \alpha_i$. In this expression, $\mathbf{x}_0$ represents the source image, while $\epsilon_t \sim \mathcal{N}(0, I)$ indicates the Gaussian noise at step $t$. In the reverse process, a denoising neural network, denoted as $D_\theta(\cdot)$, is employed to estimate the additive Gaussian noise. The optimization of this network is achieved by minimizing the loss function, $\mathcal{L}_{\text{denoise}} = \mathbb{E}_{t,\mathbf{x}_0,\epsilon \sim \mathcal{N}(0,I)}\left[\|\epsilon - D_\theta(\mathbf{x}_t, t)\|_2^2\right]$.

By employing the Bayes' theorem, it is feasible to iteratively estimate the image at timestep $t - 1$ through sampling from the posterior distribution, $p_\theta(\mathbf{x}_{t-1}|\mathbf{x}_t)$. We have $\mathbf{x}_{t-1} =$

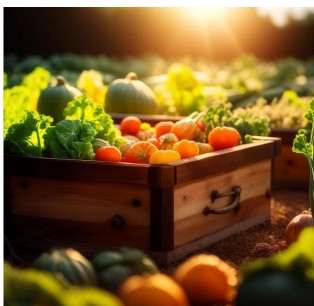

Harvest of *vegetables* in a wooden box near the *beds* vegetables grow naturally, summer light background, backlight and *sun rays*, clean sharp focus.

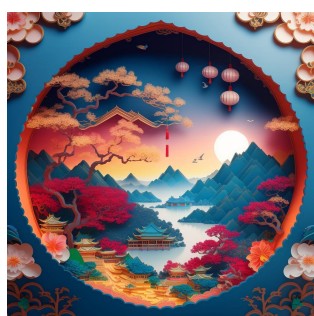

Chinese illustration, *oriental landscape painting*, above super wide angle, magical, romantic, detailed, colorful, *multi-dimensional paper kirigami craft.*

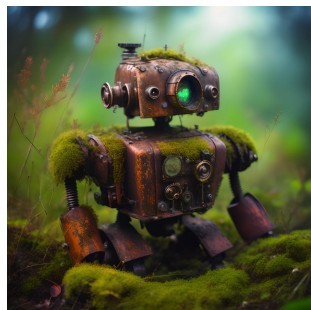

Photography closeup portrait of an adorable *rusty* broken-down *steampunk robot* covered in budding vegetation, surrounded by tall grass, *misty futuristic sci-fi forest environment.*

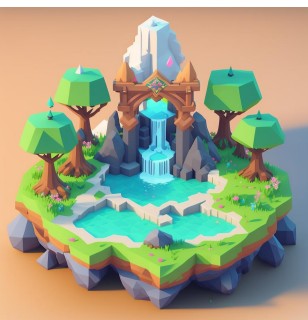

A cute little matte *low poly* isometric Zelda Breath of the *wild forest island*, *waterfalls*, soft shadows, trending on Artstation, *3d render*, monument valley, fez video game.

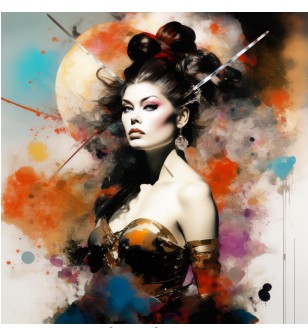

The Goddess of high fashion, impressionistic *line art*, *contrasting* earth tones, vibrant, pen and *ink* illustration, ink splatter, *abstract* expressionism superimposed onto majestic space queen.

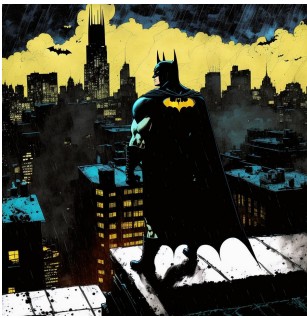

The *Caped Crusader*, Gotham *skyline,* rooftop, mysterious, powerful, *nighttime*, mixed media, expressionism, *dark tones*, high contrast, in the style of comic book artist *Frank Miller*, modern, gritty and textured, collage technique.

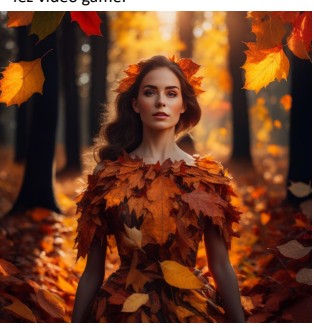

A beautiful woman dressed in a dress made of *autumn leaves* in the forest, photography, natural lighting, high detail.

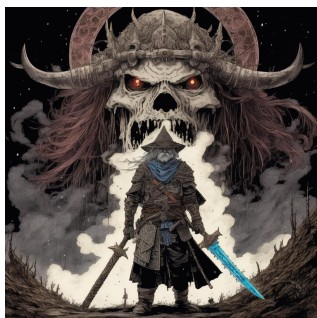

A wizard by *Q Hayashida* in the style of *Dorohedoro* for Elden Ring, with biggest most intricate *sword*, on sunlit *battlefield*, breath of the wild, striking illustration.

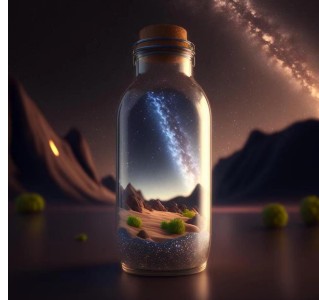

*Milkyway* in a *glass bottle*, 4k, unreal engine, octane render.

Figure 2: These examples show that RAPHAEL can generate artistic images with varying text prompts across various styles. The synthesized images have rich details and semantics. The prompts were written by human artists without cherry-picking.

$\frac{1}{\sqrt{\alpha_t}} \left( \mathbf{x}_t - \frac{1-\alpha_t}{\sqrt{1-\bar{\alpha}_t}} D_\theta \left( \mathbf{x}_t, t \right) \right) + \sigma_t z$, where $\sigma_t$ signifies the standard deviation of the newly injected noise into the image at each step, and $z$ represents the Gaussian noise.

In essence, the denoising neural network estimates the score function at varying time steps, thereby progressively recovering the structure of the image distribution. The fundamental insight provided by the DDPM lies in the fact that the perturbation of data points with noise serves to populate regions of low data density, ultimately enhancing the accuracy of estimated scores. This results in stable training and sampling.

**U-Net with Text Prompts.** The denoising network is commonly implemented using a U-Net [13] architecture, as depicted in Fig.8 in Appendix 7.3. To incorporate textual prompts (denoted by $\mathbf{y}$) into the U-Net, a text encoder neural network, $E_\theta(\mathbf{y})$, is employed to extract the textual representation.

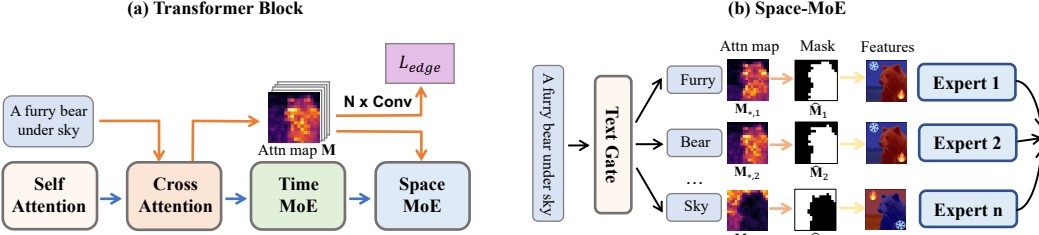

Figure 3: **Framework of RAPHAEL**. **(a)** Each block contains four primary components including a self-attention layer, a cross-attention layer, a space-MoE layer, and a time-MoE layer. The space-MoE is responsible for depicting different text concepts in specific image regions, while the time-MoE handles different diffusion timesteps. Each block uses edge-supervised cross-attention learning to further improve image quality. **(b)** shows details of space-MoE. For example, given a prompt "a furry bear under sky", each text token and its corresponding image region (given by a binary mask) are directed through distinct space experts, *i.e.,* each expert learns particular visual features at a region. By stacking several space-MoEs, we can easily learn to depict thousands of text concepts.

The extracted text tokens are input into the U-Net through a cross-attention layer. The text tokens possess a size of $n_y \times d_y$, where $n_y$ represents the number of text tokens, and $d_y$ signifies the dimension of a text token (*e.g.,* $d_y = 768$ in [14]).

The cross-attention layer can be formulated as $\mathrm{attention}(\mathbf{Q}, \mathbf{K}, \mathbf{V}) = \mathrm{softmax}\left(\frac{\mathbf{Q}\mathbf{K}^\top}{\sqrt{d}}\right)\mathbf{V}$, where $\mathbf{Q}$, $\mathbf{K}$, and $\mathbf{V}$ correspond to the query, key, and value matrices, respectively. These matrices are computed as $\mathbf{Q} = h\left(\mathbf{x}_t\right)\mathbf{W}_x^{\mathrm{qry}}$, $\mathbf{K} = E_\theta(\mathbf{y})\mathbf{W}_y^{\mathrm{key}}$, and $\mathbf{V} = E_\theta(\mathbf{y})\mathbf{W}_y^{\mathrm{val}}$, where $\mathbf{W}_x^{\mathrm{qry}} \in \mathbb{R}^{d \times d}$ and $\mathbf{W}_y^{\mathrm{key}}, \mathbf{W}_y^{\mathrm{val}} \in \mathbb{R}^{d_y \times d}$ represent the parametric projection matrices for the image and text, respectively. Additionally, $d$ denotes the dimension of an image token, $h(\mathbf{x}_t) \in \mathbb{R}^{n_x \times d}$ indicates the flattened intermediate representation within the U-Net, with $n_x$ being the number of tokens in an image. A cross-attention map between the text and image, $\mathbf{M} = \mathrm{softmax}\left(\frac{\mathbf{Q}\mathbf{K}^\top}{\sqrt{d}}\right) \in \mathbb{R}^{n_x \times n_y}$, is defined, which plays a crucial role in the proposed approach, as described in the following sections.

## 3 Our Approach

The overall framework of RAPHAEL is illustrated in Fig.3, with the network configuration details provided in the . Employing a U-Net architecture, the framework consists of 16 transformer blocks, each containing four components: a self-attention layer, a cross-attention layer, a space-MoE layer, and a time-MoE layer. The space-MoE is responsible for depicting different text concepts in specific image regions at a given scale, while the time-MoE handles different diffusion timesteps.

### 3.1 Space-MoE and Time-MoE

**Space-MoE.** Regarding the space-MoE layer, distinct text tokens correspond to various regions within an image, as previously mentioned. For instance, when provided with the prompt "a furry bear under the sky", each text token and its corresponding image region (represented by a binary mask) are fed into separate experts, as illustrated in Fig.3b. The space-MoE layer's output is the mean of all experts, calculated using the following formula: $\frac{1}{n_y} \sum_{i=1}^{n_y} e_{\mathrm{route}(\mathbf{y}_i)}\left(h'(\mathbf{x}_t) \circ \widehat{\mathbf{M}}_i\right)$. In this equation, $\widehat{\mathbf{M}}_i$ is a binary two-dimensional matrix, indicating the image region the $i$-th text token should correspond to, as shown in Fig.3b. Here, $\circ$ represents hadamard product, and $h'(\mathbf{x}_t)$ is the features from time-MoE. The gating (routing) function $\mathrm{route}(\mathbf{y}_i)$ returns the index of an expert in the space-MoE, with $\{e_1, e_2, \ldots, e_k\}$ being a set of $k$ experts.

**Text Gate Network.** The Text Gate Network is employed to distribute an image region to a specific expert, as shown in Fig.3b. The function $\mathrm{route}(\mathbf{y}_i) = \mathrm{argmax}\left(\mathrm{softmax}\left(\mathcal{G}\left(E_\theta(\mathbf{y}_i)\right) + \epsilon\right)\right)$ is used, where $\mathcal{G}: \mathbb{R}^{d_y} \mapsto \mathbb{R}^k$ is a feed forward network, which uses a text token representation $E_\theta(\mathbf{y}_i)$ as input and assigns a space expert. To prevent mode collapse, random noise $\epsilon$ is incorporated. The

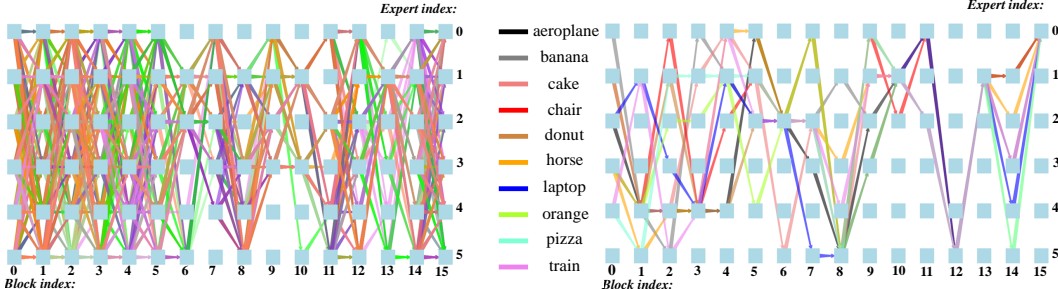

Figure 4: **Left:** We visualize the diffusion paths (routes) from the network input to the output, utilizing 16 space-MoE layers, each containing 6 spatial experts. These paths are closely associated with 100 adjectives, such as "scenic", "peaceful", and "majestic", which represent the most frequently occurring adjectives for describing artworks as suggested by GPT-3.5 [15, 16]. Given that GPT-3.5 has been trained on trillions of tokens, we believe that these adjectives reflect a diverse, real-world distribution. Our findings indicate that different paths distinctively represent various adjectives. **Right:** We depict the diffusion paths for ten categories (*i.e.,* nouns) within the COCO dataset. Our observations reveal that different categories activate distinct paths in a heterogeneous manner. The display colors blend together where the routes overlap.

`argmax` function ensures that one expert exclusively handles the corresponding image region for each text token, without increasing computational complexity.

**From Text to Image Region.** Recall that $\mathbf{M}$ is the cross-attention map between text and image, where each element, $\mathbf{M}_{j,i}$, represents a correspondence value between the $j$-th image token and the $i$-th text token. In the space-MoE, each entry in the binary mask $\widehat{\mathbf{M}}_i$ equals "1" if $\mathbf{M}_{j,i} \geq \eta_i$, otherwise "0" if $\mathbf{M}_{j,i} < \eta_i$, as illustrated in Fig.3b. A thresholding mechanism is introduced to determine the values in the mask. The threshold value $\eta_i = \alpha \max(\mathbf{M}_{*,i})$ is defined, where $\max(\mathbf{M}_{*,i})$ represents the maximum correspondence between text token $i$ and all image regions. The hyper-parameter $\alpha$ will be evaluated through an ablation study.

**Discussions.** The insight behind the space-MoE is to effectively model the intricate relationships between text tokens and their corresponding regions in the image, accurately reflecting concepts in the generated images. As illustrated in Fig.4, the employment of 16 space-MoE layers, each containing 6 experts, results in billions of spatial diffusion paths (*i.e.,* $6^{16}$ possible routes). It is evident that each diffusion path is closely associated with a specific textual concept.

To investigate this further, we generate 100 prevalent adjectives that are the most frequently occurring adjectives for describing artworks as suggested by GPT-3.5 [15, 16]. Given that GPT-3.5 has been trained on trillions of tokens, we posit that these adjectives reflect a diverse, real-world distribution. We input each adjective into the RAPHAEL model to generate 100 distinct images and collect their corresponding diffusion paths. Consequently, we obtain ten thousand paths for the 100 words. By treating these pathways as features (*i.e.,* each path is a vector of 16 entries), we train a straightforward classifier (*e.g.,* XGBoost [17]) to categorize the words. The classifier after 5-fold cross-validation achieves over 93% accuracy for open-world adjectives, demonstrating that different diffusion paths distinctively represent various textual concepts. We observe analogous phenomena within the 80 object categories of the COCO dataset. Further details on verbs and visualization are provided in the Appendix 7.5.

**Time-MoE.** We can further enhance the image quality by employing a time-mixture-of-experts (time-MoE) approach, which is inspired by previous works such as [4, 5]. Given that the diffusion process iteratively corrupts an image with Gaussian noise over a series of timesteps $t = 1, \dots, T$, the image generator is trained to denoise the images in reverse order from $t = T$ to $t = 1$. All timesteps aim to denoise a noisy image, progressively transforming random noise into an artistic image. Intuitively, the difficulty of these denoising steps varies depending on the noise ratio presented in the image. For example, when $t = T$, the denoising network's input image $\mathbf{x}_t$ is highly noisy. When $t = 1$, the image $\mathbf{x}_t$ is closer to the original image.

To address this issue, we employ a time-MoE before each space-MoE in each transformer block. In contrast to [4, 5] , which necessitate hand-crafted time expert assignments, we implement an additional gate network to automatically learn to assign different timesteps to various time experts. Further details can be found in the Appendix 7.3.

## 3.2 Edge-supervised Learning

In order to further enhance the image quality, we propose incorporating an edge-supervised learning strategy to train the transformer block. By implementing an edge detection module, we aim to extract rich boundary information from an image. These intricate boundaries can serve as supervision to guide the model in preserving detailed image features across various styles.

Consider a neural network module, $P_\theta(\mathbf{M})$, with parameters of $N$ convolutional layers (*e.g.,* $N = 5$). This module is designed to predict an edge map given an attention map $\mathbf{M}$ (refer to Fig.7a in the Appendix 7.2). We utilize the edge map of the input image, denoted as $\mathbf{I}_{edge}$, to supervise the network $P_\theta$. $\mathbf{I}_{edge}$ can be obtained by the holistically-nested edge detection algorithm [18] (Fig.7b). Intuitively, the network $P_\theta$ can be trained by minimizing the loss function, $\mathcal{L}_{edge} = \text{Focal}(P_\theta(\mathbf{M}), \mathbf{I}_{edge})$, where $\text{Focal}(\cdot, \cdot)$ denotes the focal loss [19] employed to measure the discrepancy between the predicted and the "ground-truth" edge maps. Moreover, as discussed in [5, 6], the attention map $\mathbf{M}$ is prone to becoming vague when the timestep $t$ is large. Consequently, it is essential to adopt a timestep threshold value to inactivate (pause) edge-supervised learning when $t$ is large. This timestep threshold value ($T_c$) is a hyper-parameter that will be evaluated through an ablation study.

Overall, the RAPHAEL model is trained by combining two loss functions, $\mathcal{L} = \mathcal{L}_{denoise} + \mathcal{L}_{edge}$. As demonstrated in Fig.7d in the Appendix 7.2, edge-supervised learning substantially improves the image quality and aesthetic appeal of the generated images.

# 4 Experiments

This section presents the experimental setups, the quantitative results compared to recent state-of-the-art models, and the ablation study to demonstrate the effectiveness of RAPHAEL. More artistic images generated by RAPHAEL and comparisons between RAPHAEL and other diffusion models can be found in Appendix 7.6 and 7.7.

**Dataset**. The training dataset consists of a subset of LAION-5B [20] and some internal datasets, including 730M text-images pairs in total. To collect training data from LAION-5B, we filter the images using the aesthetic scorer same as Stable Diffusion [2] and remove the image-text pairs that have scores smaller than $4.7$. We remove the images with watermarks either. Since the text descriptions in LAION-5B are noisy, we clean them by removing useless information such as URLs, HTML tags, and email addresses, inspired by [2, 4, 21].

**Multi-scale Training**. To improve text-image alignment, instead of cropping images to a fixed scale [2], we resize an image to its nearest size into different buckets, which has 9 different image scales. Additionally, the GPU resources will be automatically allocated to each bucket depending on the number of images it contains, enabling effective use of computational resources*.

**Implementations**. To reduce training and sampling complexity, we use a Variational Autoencoder (VAE) [22, 23] to compress images using Latent Diffusion Model [2]. We first pre-train an image encoder to transform an image from pixel space to a latent space, and an image decoder to convert it back. Unlike previous works, the cross-attention layers in RAPHAEL are augmented with space-MoE and time-MoE layers. The entire model is implemented in PyTorch [24], and is trained by AdamW [25] optimizer with a learning rate of $1e - 4$, a weight decay of $0$, a batch size of $2,000$, on $1,000$ NVIDIA A100s for two months. More details on the hyper-parameter settings can be found in the Appendix 7.1.

## 4.1 Comparisons

**Results on COCO**. Following previous works [1, 2, 4], we evaluate RAPHAEL on the COCO $256 \times 256$ dataset using zero-shot Frechet Inception Distance (FID), which measures the quality and diversity of images. Similar to [1, 2, 4, 5, 32], $30,000$ images are randomly selected from the validation set for evaluation. Table 1 shows that RAPHAEL achieves a new state-of-the-art

---

*The dimensions of each bucket are as follows: [448, 832], [512, 768], [512, 704], [640, 640], [576, 640], [640, 576], [704, 512], [768, 512], and [832, 448]. For instance, when images are resized, those with an aspect ratio of 1.0 will be assigned to the bucket of size [640, 640]. GPUs will be allocated to each bucket, based on the images it contains. All GPUs will have the same batch size and will select images from its associated bucket.

Table 1: **Comparisons** of RAPHAEL with the recent representative text-to-image generation models on the MS-COCO $256 \times 256$ using zero-shot FID-30k. We see that RAPHAEL outperforms all previous works in image quality, even a commercial product released recently.

| Approach | Venue/Date | Model Type | FID-30K | Zero-shot FID-30K |
|---|---|---|---|---|
| DF-GAN [26] | CVPR'22 | GAN | 21.42 | - |
| DM-GAN + CL [27] | CVPR'19 | GAN | 20.79 | - |
| LAFITE [28] | CVPR'22 | GAN | 8.12 | - |
| Make-A-Scene [29] | ECCV'22 | Autoregressive | 7.55 | - |
| LDM [2] | CVPR'22 | Diffusion | - | 12.63 |
| GLIDE [30] | ICML'22 | Diffusion | - | 12.24 |
| DALL-E 2 [3] | arXiv, April 2022 | Diffusion | - | 10.39 |
| GigaGAN [31] | CVPR'23 | GAN | - | 9.09 |
| Stable Diffusion [2] | CVPR'22 | Diffusion | - | 8.32 |
| Muse-3B [32] | arXiv, Jan. 2023 | Non-Autoregressive | - | 7.88 |
| Imagen [1] | NeurIPS'22 | Diffusion | - | 7.27 |
| eDiff-I [4] | arXiv, Nov. 2022 | Diffusion Experts | - | 6.95 |
| ERNIE-ViLG 2.0 [5] | CVPR'23 | Diffusion Experts | - | 6.75 |
| DeepFloyd | Product, May 2023 | Diffusion | - | 6.66 |
| RAPHAEL | - | Diffusion Experts | - | **6.61** |

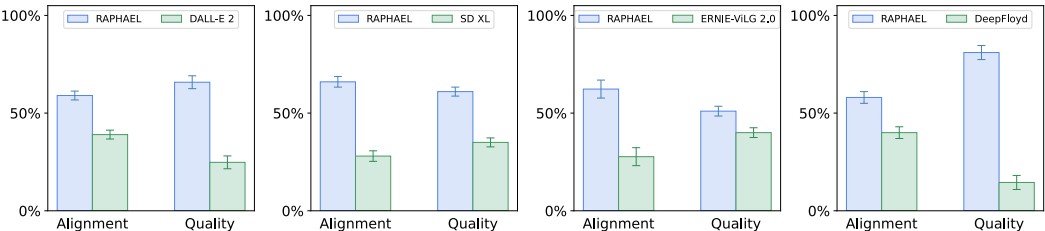

Figure 5: **Comparisons** of RAPHAEL with DALL-E 2, Stable Diffusion XL (SD XL), ERNIE-ViLG 2.0, and DeepFloyd in a user study using the ViLG-300 benchmark. We report the user's preference rates with 95% confidence intervals. We see that RAPHAEL can generate images with higher quality and better conform to the prompts.

performance of text-to-image generation, with 6.61 zero-shot FID-30k on MS-COCO, surpassing prominent image generators such as Stable Diffusion, Imagen, ERNIE-ViLG 2.0, and DALL-E 2.

**Human Evaluations**. We employ the ViLG-300 benchmark [5], a bilingual prompt set, which enables to systematically evaluate text-to-image models given various text prompts in Chinese and English. ViLG-300 allows us to convincingly compare RAPHAEL with recent-advanced models including DALL-E 2, Stable Diffusion, ERNIE-ViLG 2.0, and DeepFloyd, in terms of both image quality and text-image alignment. For example, human artists are presented with two sets of images generated by RAPHAEL and a competitor, respectively. They are asked to compare these images from two aspects respectively, including image-text alignment, and image quality and aesthetics. Throughout the entire process, human artists are unaware of which model the image is generated from. Fig.5 shows that RAPHAEL surpasses all other models in both image-text alignment and image quality in the user study, indicating that RAPHAEL can generate high-artistry images that conform to the text.

**Extensions to LoRA, ControlNet, and SR-GAN.** RAPHAEL can be further extended by incorporating LoRA, ControlNet, and SR-GAN. In Appendix 7.8, we present a comparison between RAPHAEL and Stable Diffusion utilizing LoRA. RAPHAEL demonstrates superior robustness against overfitting compared to Stable Diffusion. We also demonstrate RAPHAEL with a canny-based ControlNet. Furthermore, by employing a tailormade SR-GAN model, we enhance the image resolution to $4096 \times 6144$.

## 4.2 Ablation Study

**Evaluate every module in RAPHAEL.** We conduct a comprehensive assessment of each module within the RAPHAEL model, utilizing the CLIP [14] score to measure image-text alignment. Given the significance of classifier-free guidance weight in controlling image quality and text alignment, we present ablation results as trade-off curves between CLIP and FID scores across a range of

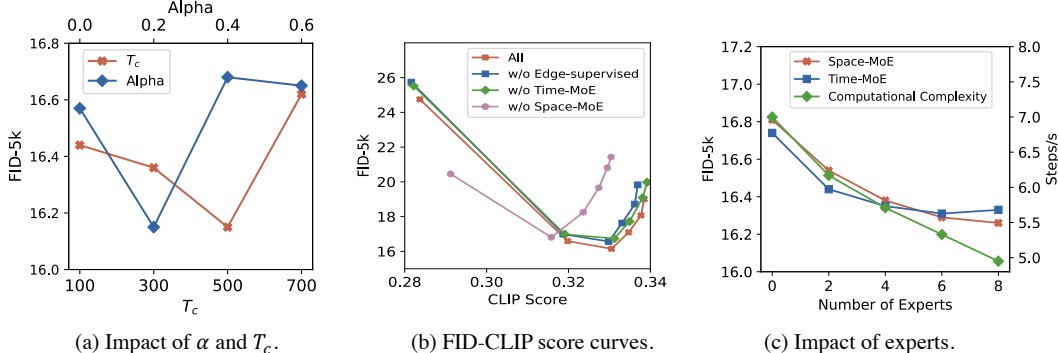

Figure 6: **Ablation Study**. (a) examines the selection of $\alpha$ and $T_c$. (b) presents the trade-off between FID and CLIP scores for the complete RAPHAEL model and its variants without space-MoE, time-MoE, and edge-supervised learning. (c) visualizes the correlation between FID-5k and runtime complexity (measured in terms of the number of DDIM [34] steps for an image per second) as a function of the number of experts employed. Notably, the computational complexity is predominantly influenced by the number of spatial experts.

guidance weights [33], specifically $1.5, 3.0, 4.5, 6.0, 7.5$, and $9.0$. Fig.6b compares these curves for the complete RAPHAEL model and its variants without space-MoE, edge-supervised learning, and time-MoE, respectively. Our findings indicate that all modules contribute effectively. For example, space-MoE substantially enhances the CLIP score and the optimal guidance weight for the sampler shifts from 3.0 to 4.5. Moreover, at the same guidance weight, space-MoE considerably reduces the FID, resulting in a significant improvement in image quality.

**Choice of $\alpha$ and $T_c$.** As depicted in Fig.6a, we observe that $\alpha = 0.2$ delivers the best performance, implying a balance between preserving adequate features and avoiding the use of the entire latent features. An appropriate threshold value for $T_c$ terminates edge-supervised learning when the diffusion timestep is large. Our experiments reveal that a suitable choice for $T_c$ is 500, ensuring the effective learning of texture information.

**Performance and Runtime Analysis on Number of Experts.** We offer an examination of the number of experts, ranging from $0$ to $8$, in Fig.6c. For each setting, we employ 100 million training samples. Our results demonstrate that increasing the number of experts improves FID (lower values are preferable). However, adding spatial experts introduces additional computations, with the computational complexity bounded by the total number of experts. Once all available experts have been deployed, the computational complexity ceases to grow. In the right-hand side of Fig.6c, we provide a runtime analysis for 40 input tokens, ensuring the utilization of all space experts. For instance, when the number of experts is 6, the inference speed decreases by 24% but yields superior fidelity. This remains faster than previous diffusion models such as Imagen [1] and eDiff-I [4].

## 5 Related Work

We review related works from two perspectives, mixture-of-experts and text-to-image generation. More related works can be found in Appendix 7.4. Firstly, the Mixture-of-Experts (MoE) method [7, 8] partitions model parameters into distinct subsets, each termed an "expert". The MoE paradigm finds applicability beyond language processing tasks, extending to visual models [35] and Mixture-of-Modality-Experts within multi-modal transformers [36]. Additionally, efforts are being made to accelerate the training or inference processes for MoE [37, 38]. Secondly, text-to-image generation is to synthesize images from natural language descriptions. Early approaches relied on generative adversarial networks (GANs) [39, 40, 41, 42] to generate images. More recently, with the transformative success of transformers in generative tasks, models such as DALL-E [43], Cogview [44], and Make-A-Scene [29] have treated text-to-image generation as a sequence-to-sequence problem, utilizing auto-regressive transformers as generators and employing text/image tokens as input/output sequences. Recently, another research direction has focused on diffusion models by integrating textual conditioning within denoising steps, like Stable Diffusion [2], DALL-E 2 [3], eDiff-I [4], ERNIE-ViLG 2.0 [5], and Imagen [1].

# 6 Conclusion

This paper introduces RAPHAEL, a novel text-conditional image diffusion model capable of generating highly-artistic images using a large-scale mixture of diffusion paths. We carefully design space-MoE and time-MoE within an edge-supervised learning framework, enabling RAPHAEL to accurately portray text prompts, enhance the alignment between textual concepts and image regions, and produce images with superior aesthetic appeal. Comprehensive experiments demonstrate that RAPHAEL surpasses previous approaches, such as Stable Diffusion, ERNIE-ViLG 2.0, DeepFloyd, and DALL-E 2, in both FID-30k and the human evaluation benchmark ViLG-300. Additionally, RAPHAEL can be extended using LoRA, ControlNet, and SR-GAN. We believe that RAPHAEL has the potential to advance image generation research in both academia and industry.

**Limitation and Potential Negative Societal Impact.** We acknowledge some limitations in our paper that require attention. One limitation is the direct binarization of the attention map, which may result in the loss of some information. An adaptive module should be proposed to address this issue effectively. Additionally, the performance may be affected by failure cases of the edge detector, leading to potential degradation. We plan to explore solutions for these limitations in our future work. The potential negative social impact is to use the RAPHAEL API to create images containing misleading or false information. This issue potentially presents in all powerful text-to-image generators. We will solve this issue (*e.g.,* by prompt filtering) before releasing the API to the public.

## Acknowledgments and Disclosure of Funding

This paper is partially supported by the National Key R&D Program of China No.2022ZD0161000 and the General Research Fund of Hong Kong No.17200622.

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
