# 7 Appendix

## 7.1 Hyper-parameters and Values

We give the hyper-parameters and values in Table 2.

Table 2: Hyper-parameters and values in RAPHAEL.

| Configs/Hyper-parameters | Values |
| --- | --- |
| $T$ | 1000 |
| $n_y$ | 77 |
| $d_y$ | 1024 |
| $T_c$ | 500 |
| $\alpha$ | 0.2 |
| Betas of AdamW [25] | (0.9, 0.999) |
| Weight decay | 0.0 |
| Learning rate | $1e$-4 |
| Number of space experts | 6 |
| Number of time experts | 4 |
| Warmup steps | 20000 |
| Batch size | 2000 |
| Number of GPUs | 1000 |
| Number of transformer blocks | 16 |
| Use checkpoint | True |
| $\alpha$ in Focal Loss [19] | 0.5 |
| $\gamma$ in Focal Loss [19] | 2 |
| Text encoder | OpenCLIP-g/14 [14] |
| Enable multi-scale training | True |
| Activations in experts and gate network | GELU |
| Architectures of experts and gate network | FFN |

## 7.2 Details of Edge-supervised Learning

We provide some demonstrations of edge-supervised learning in Fig.7. The reason to choose Focal loss is when we adopt cross-entropy loss, we encounter imbalanced edge-maps and background, where the prediction module tends to classify all pixels as background, negatively impacting the cross-attention maps.

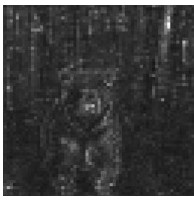 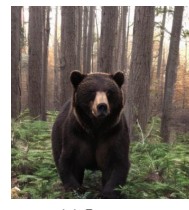 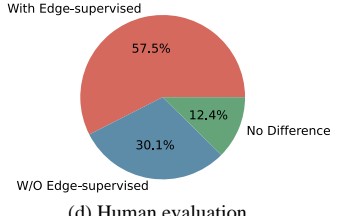

|         (a) Attention map.         |         (b) Edge map.         |         (c) Image.         |         (d) Human evaluation.         |

Figure 7: From left to right, we display the attention map corresponding to the pooled token in CLIP, the ground truth edges identified by the edge detection algorithm, and the associated image. In the fourth figure, we present the human evaluation results for models with and without edge-supervised learning on the ViLG-300 benchmark. Evaluators are instructed to compare these images considering image aesthetics and we report the preference rates, and our findings indicate that edge-supervised learning significantly enhances the aesthetic quality of the images.

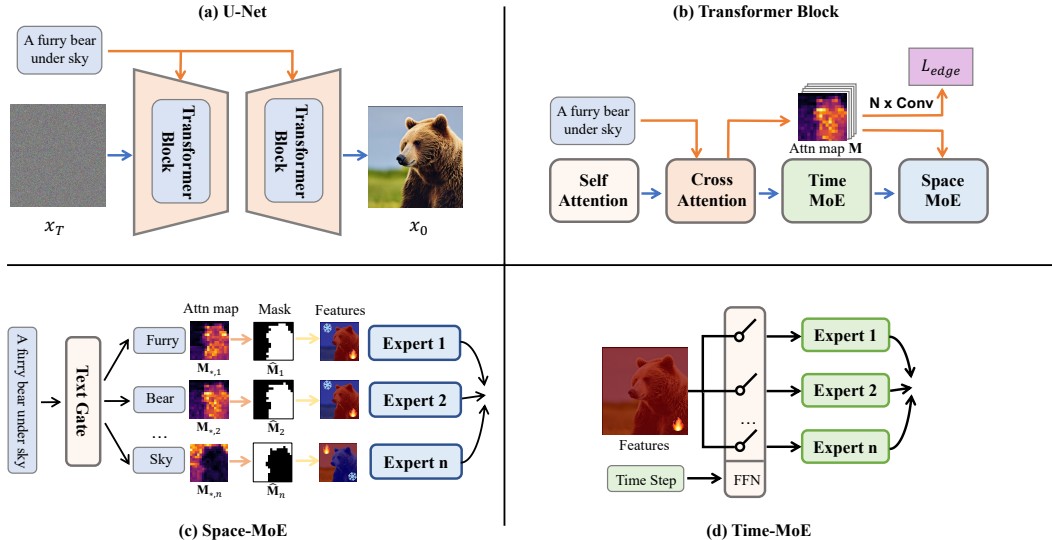

Figure 8: **(a)** The architecture of U-Net, which consists of many transformer blocks. **(b)** Each block contains four primary components including a self-attention layer, a cross-attention layer, a space-MoE layer, and a time-MoE layer. The space-MoE is responsible for depicting different text concepts in specific image regions, while the time-MoE handles different diffusion timesteps. Each block uses edge-supervised cross-attention learning to further improve image quality. **(c)** shows details of space-MoE. For example, given a prompt "a furry bear under sky", each text token and its corresponding image region (given by a binary mask) are directed through distinct space experts, *i.e.,* each expert learns particular visual features at a region. **(d)** For time-MoE, an initial timestep is provided, followed by the selection of an expert responsible for handling the visual features.

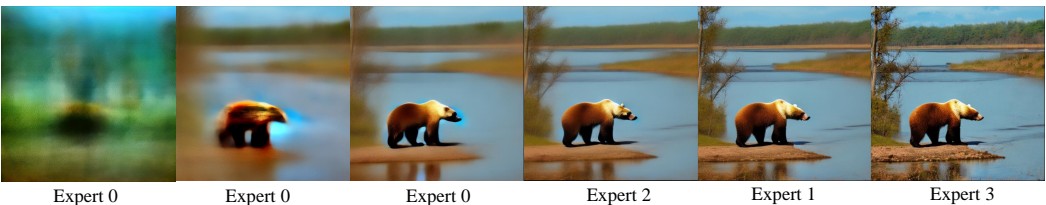

Figure 9: The routes of time-MoE in the first transformer block, where the first expert focuses on noisy images, while other experts handle images with low noise levels.

## 7.3 Details on Time-MoE

The overall architecture of RAPHAEL can be found in Fig.8. The Time-MoE is composed of a Time Gate Network to distribute the features to a specific expert according to the timestep, which can be formulated as $h'(\mathbf{x}_t) = te_{\text{t\_router}(t_i)}(h_c(\mathbf{x}_t))$. In this equation, $h_c(\mathbf{x}_t)$ is the features from cross-attention module. The gating function t\_router returns the index of an expert in the Time-MoE, with $\{te_1, te_2, ..., te_{n_t}\}$ being a set of $n_t$ experts. Concretely, the Time Gate Network is implemented by a function, $\text{t\_router}(t_i) = \text{argmax}\left(\text{softmax}\left(\mathcal{G}'\left(E'_\theta(t_i)\right) + \epsilon\right)\right)$ at timestep $t_i$. To prevent mode collapse, random noise $\epsilon$ is incorporated. Similar to the Text Gate, $\mathcal{G}' : \mathbb{R}^{d_t} \mapsto \mathbb{R}^{n_t}$, is a feed forward network, where $d_t$ is the dimension of the time embedding $E'_\theta(t_i)$.

**Analysis.** In our exploration, we uncover some statistical regularities within the routes of time experts across all transformer blocks, establishing a clear correlation with the timestep dimension. Notably, we observe a distinct division of labor among these experts, specializing in timesteps characterized by varying levels of noise. For instance, as illustrated in Fig.9, in the first transformer block, the first expert predominately focuses on processing noisy images (representing the initial 59% of DDIM sampler steps), while the remaining experts handle images with relatively lower noise levels (representing the final 41% of DDIM sampler steps). This systematic allocation of expertise based on noise characteristics underscores the model's ability to adapt its computational resources efficiently and effectively.

## 7.4 Related Work

Foundation models have achieved remarkable success in several fields [2, 45, 35, 15], especially for text-to-image generation. We review related works from two perspectives, mixture-of-experts, and text-to-image generation.

**Mixture-of-Experts.** The Mixture-of-Experts (MoE) method [7, 8] in neural networks partitions specific model parameters into distinct subsets, each termed an "expert." During forward propagation, a dynamic routing mechanism assigns these experts to diverse inputs, with each input exclusively interacting with its selected experts. MoE models implement a learned gating function that selectively activates a subset of experts, enabling the input to engage either all experts [46] or a sparse mixture thereof [8, 47], as evidenced in recent expansive language models. While a multitude of models employs experts strictly within the linear layers, other research regards an entire language model as an expert [48]. The MoE paradigm finds applicability beyond language processing tasks, extending to visual models [35] and Mixture-of-Modality-Experts within multi-modal transformers [36]. Additionally, efforts are being made to accelerate the training or inference processes within the MoE paradigm [37, 38].

**Text-to-Image Generation.** Text-to-image generation, the task of synthesizing images from natural language descriptions, has experienced significant progress in recent years. Early approaches relied on generative adversarial networks (GANs) [39, 40, 41, 42, 31] to generate images. More recently, with the transformative success of transformers in generative tasks, models such as DALL-E [43], Cogview [44], and Make-A-Scene [29] have treated text-to-image generation as a sequence-to-sequence problem, utilizing auto-regressive transformers as generators and employing text/image tokens as input/output sequences. Recently, another research direction has focused on diffusion models, framing the task as an iterative denoising process. By integrating textual conditioning within denoising steps, models like Stable Diffusion [2], DALL-E 2 [3], eDiff-I [4], ERNIE-ViLG 2.0 [5], and Imagen [1] have consistently set new benchmarks in text-to-image generation. Specifically, Stable Diffusion and ERNIE-ViLG 2.0 map images into a latent space, following the Latent Diffusion Model paradigm to enhance training and sampling efficiency, while DALL-E 2, eDiff-I, and Imagen operate in pixel space. Furthermore, diffusion models also show great potential in image editing [49, 6, 50, 51], personalized generation [52, 53, 54, 55], and 3D/video/gesture generation [56, 57, 58, 59, 60, 61]. ControlNet [10, 62] is a noteworthy model in the text-to-image generation landscape. It builds upon the concept of controllable image synthesis, wherein generated images can be manipulated based on user-defined constraints or attributes.

## 7.5 More Details on Routers of Space-MoE

We continue to delve into the diffusion paths of both COCO categories and verbs, uncovering intriguing insights. Utilizing the powerful GPT-3.5 [15, 16], we randomly generate 50 verbs to enrich our investigation. Moreover, employing the prompt template randomly generated by GPT-3.5, we generate 100 samples for each COCO category and verb [15, 16]. Similar to Section 3.1, by adopting XGBoost as our classifier, we find that the accuracy rate reaches 94.3% and 97.5%, respectively. We give the routes of COCO and 50 verbs in Fig.10. We also show more visualization results of the attention maps in Fig.11. We provide the adjectives and verbs in the following section. Moreover, we observe similar concepts exhibit similar diffusion paths, as shown in Fig. 12.

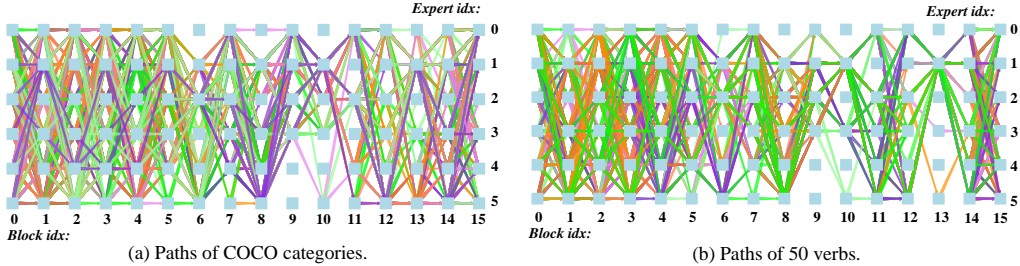

(a) Paths of COCO categories.

(b) Paths of 50 verbs.

Figure 10: We visualize the diffusion paths (routes) from the network input to the output, utilizing 16 space-MoE layers, each containing 6 space experts. These paths are closely associated with COCO categories and 50 verbs.

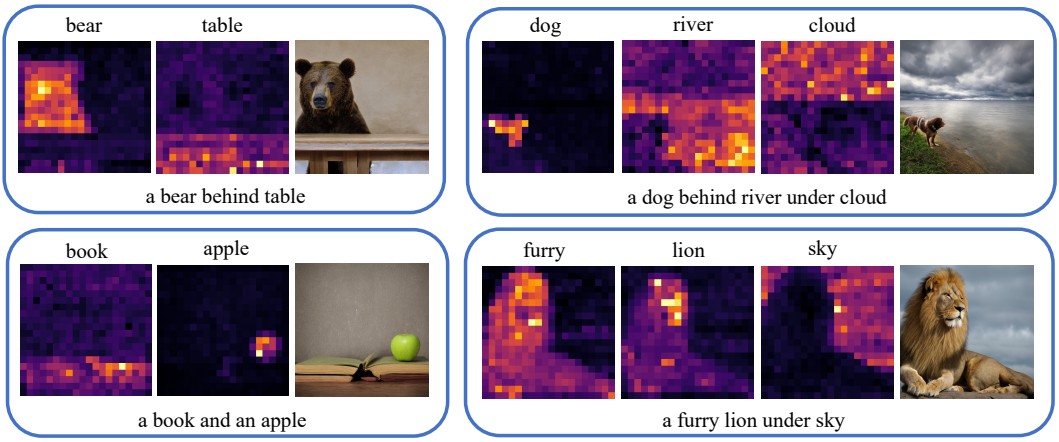

Figure 11: We give the prompts, their associated generated images, and attention maps.

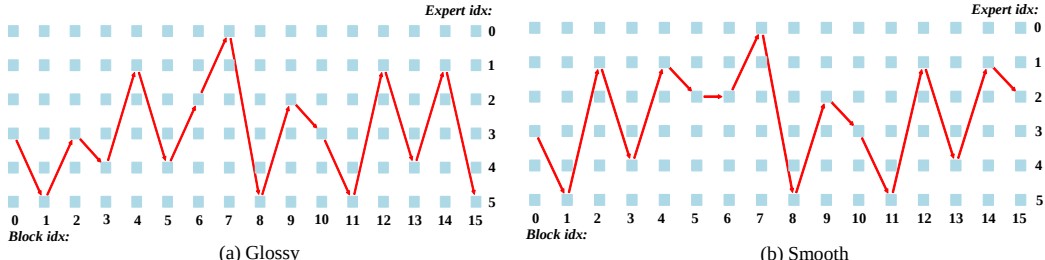

(a) Glossy

(b) Smooth

Figure 12: Similar concepts exhibit similar diffusion paths, such as glossy and smooth.

### 7.5.1 Adjectives and Verbs

**Adjectives.** Aesthetic, alluring, artistic, astonishing, attractive, baroque, beautiful, blissful, captivating, chic, classic, coastal, colorful, common, dark, decorative, delicate, dramatic, dreamlike, dreamy, dynamic, eclectic, elegant, emotive, enchanting, energetic, enthralling, essential, ethereal, evocative, extraordinary, fascinating, flexible, fragile, futuristic, glamorous, glossy, gorgeous, gothic, grand, harmonious, idyllic, impressive, industrial, innovative, inspiring, intricate, intriguing, joyful, lively, luxurious, magnificent, meditative, mesmerizing, minimal, minimalist, modern, moroccan, mysterious, nostalgic, ordinary, patterned, peaceful, picturesque, plain, playful, practical, quirky, rare, renaissance, retro, rigid, romantic, rough, rustic, satisfying, Scandinavian, scenic, serene, serious, shiny, simple, sleek, smooth, sophisticated, static, striking, stunning, sturdy, stylish, textured, traditional, tranquil, unique, unusual, useful, vibrant, victorian, vivid, whimsical.

**Verbs.** Balance, blend, blossom, bond, carve, celebrate, cheer, climb, collaborate, conduct, conquer, cook, craft, create, dance, dream, embrace, experiment, explore, gaze, harmonize, hike, hug, ignite, illuminate, jump, laugh, leap, listen, meander, observe, paint, play, ponder, read, rejoice, relax, ride, run, savor, sculpt, sing, smile, soar, surf, swim, swing, taste, wander, whisper.

### 7.6 More Comparisons between RAPHAEL and Prestigious Diffusion Models

In this section, we provide more comparisons between RAPHAEL and Midjourney, Stable Diffusion XL, DALL-E 2, DeepFloyd, ERNIE-ViLG 2.0 in Fig.13 and 14.

**DALL-E 2**

**Midjourney V5.1**

**Stable Diffusion XL**

**ERNIE ViLG 2.0**

**DeepFloyd**

**RAPHAEL**

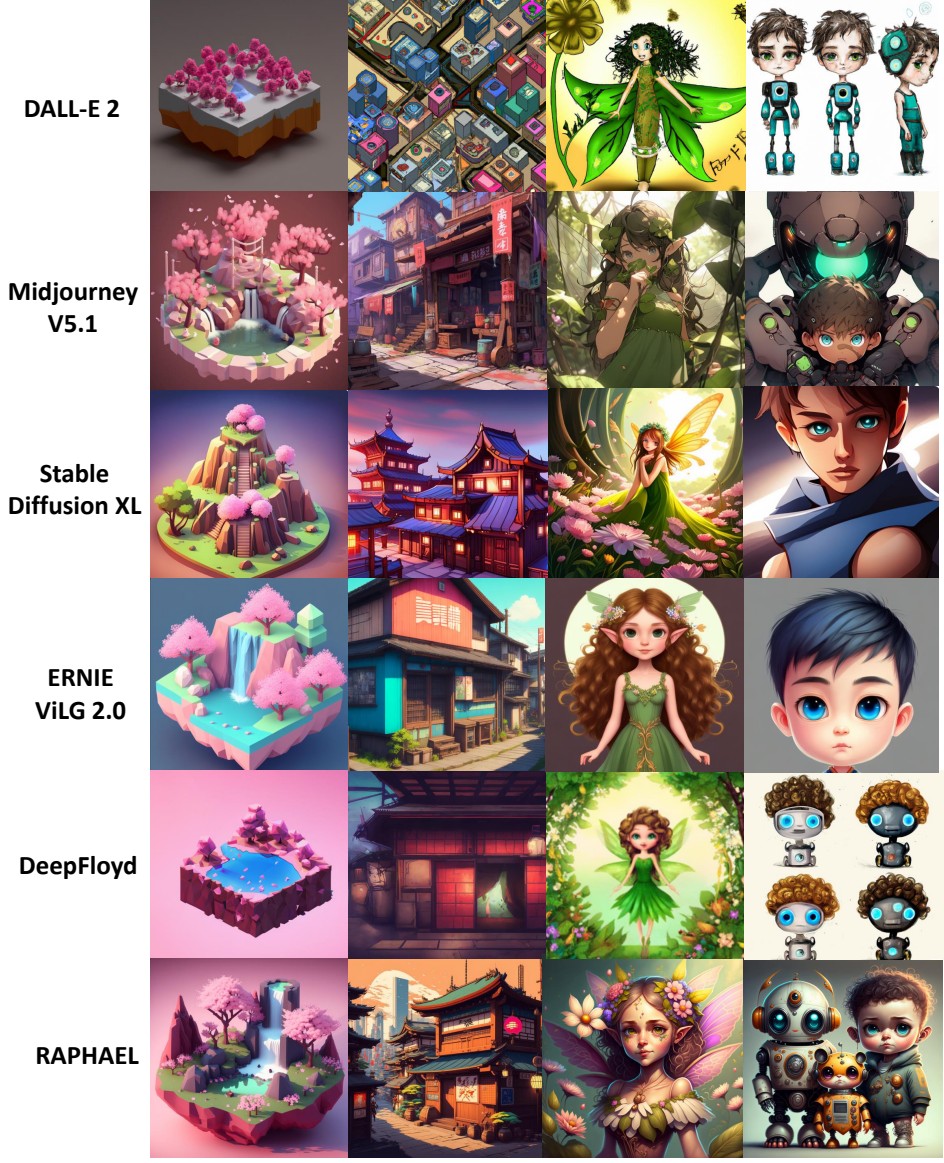

1. A cute little matte low poly isometric *cherry blossom forest island*, *waterfalls*, lighting, soft shadows, trending on Artstation, 3d render, monument valley, fez video game.

2. A shanty version of Tokyo, new rustic style, *bold colors with all colors palette*, video game, genshin, tribe, fantasy, overwatch.

3. Cartoon characters, mini characters, figures, illustrations, flower fairy, green dress, *brown hair, curly long hair, elf-like wings, many flowers and leaves*, natural scenery, *golden eyes*, detailed light and shadow , a high degree of detail.

4. Cartoon characters, mini characters, hand-made, illustrations, *robot kids*, color expressions, boy, *short brown hair, curly hair*, *blue eyes*, technological age, *cyberpunk*, big eyes, cute, mini, detailed light and shadow, high detail.

Figure 13: The prompts for each column are given in the figure. We give the comparisons between DALL-E 2 Midjourney v5.1, Stable Diffusion XL, ERNIE ViLG 2.0, DeepFloyd, and RAPHAEL. They are given the same prompts, where the words that the human artists yearn to preserve within the generated images are highlighted in red. Only the RAPHAEL-generated images precisely reflect the prompts, while other models generate compromised results. For images with cartoon styles, we switch Midjourney v5.1 to Nijijourney v5.

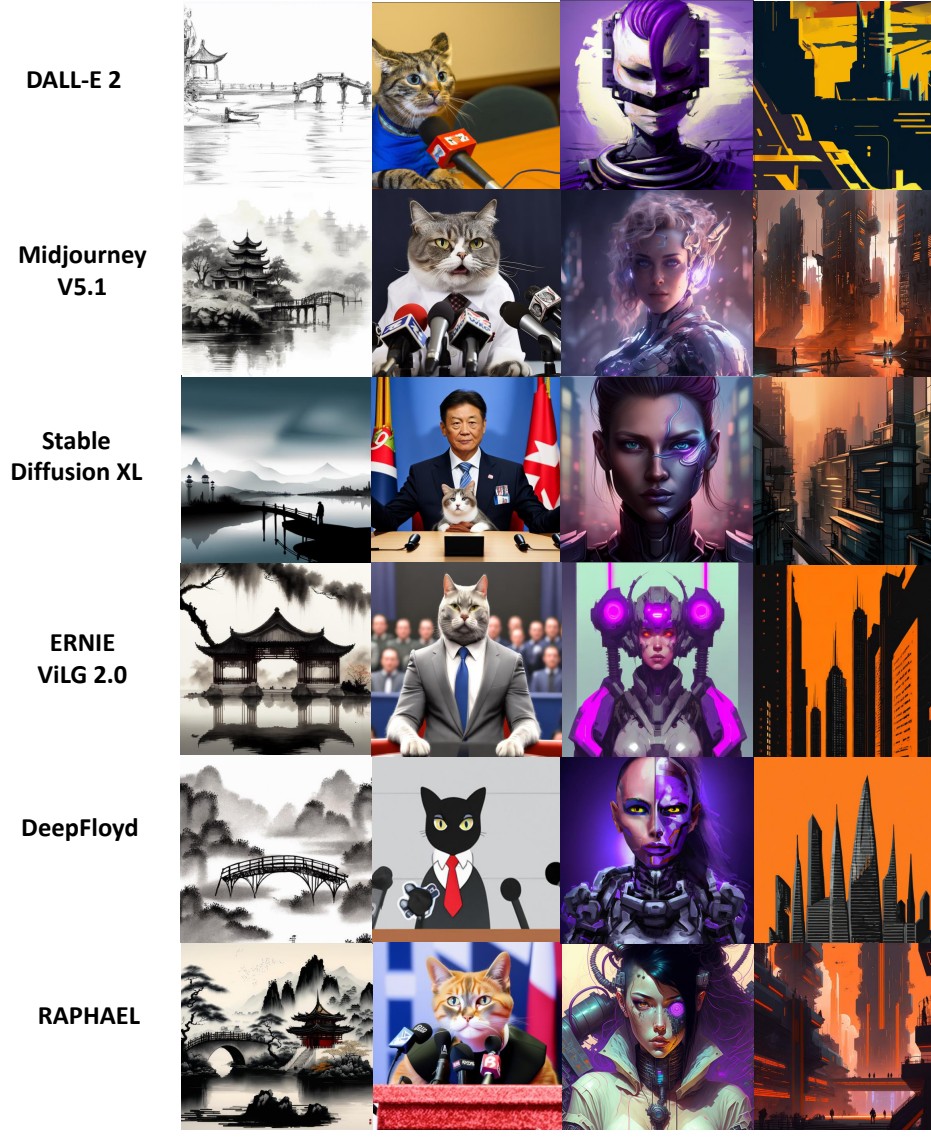

DALL-E 2

Midjourney V5.1

Stable Diffusion XL

ERNIE ViLG 2.0

DeepFloyd

RAPHAEL

1. Landscape, *lake, buildings, bridge*, Chinese ink style.

2. Photo of *an athlete cat* explaining it's latest scandal at a *press conference* to journalists.

3. Impasto, illustration character, illustration poster, techno, cyberpunk*, half human, half robot, repaired human*, human warrior, mech display, woman in mech, purple eyes, *half face destroyed*, trend on Artstation, high detail, detailed light and shadow, 4k.

4. Atmosphere, flat painting illustration, illustration, *unique tall buildings in the city*, irregular design, iron age, *black and orange*, detailed light and shadow, cyberpunk, technological age, high detail, *CG feeling.*

Figure 14: The prompts for each column are given in the figure. We give the comparisons between DALL-E 2 Midjourney v5.1, Stable Diffusion XL, ERNIE ViLG 2.0, DeepFloyd, and RAPHAEL. They are given the same prompts, where the words that the human artists yearn to preserve within the generated images are highlighted in red. Only the RAPHAEL-generated images precisely reflect the prompts, while other models generate compromised results.

## 7.7 More Images Generated by RAPHAEL

We give more cases in Fig.15, 16, 17, 18, and 19.

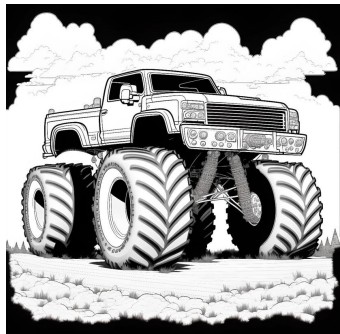

Coloring page for kids,Giant Monster Truck Lifted With Big Wheels,cartoon style, thick lines, low detail.

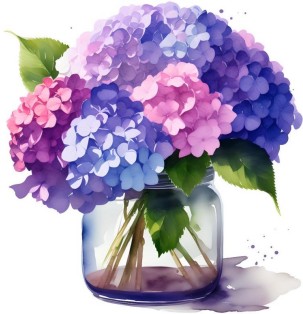

Watercolor purple Hydrangea clipart Hydrangea floral bouquet in a jar, white background, no text.

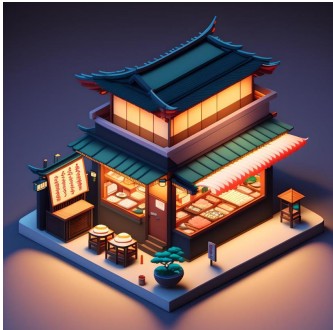

Sushi shop, isometric, high quality, old tokyo vibes, night, Japanese, 3d.

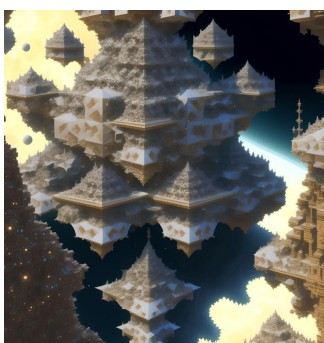

Spacefaring civilization silicon-based lifeforms megastructures built into fractal mineralized shapes, Menger sponge Sierpinski gasket koch snowflake Mandelbulb, key visual by craig mullins and yoji Ishikawa in the style of fractal core spacewave.

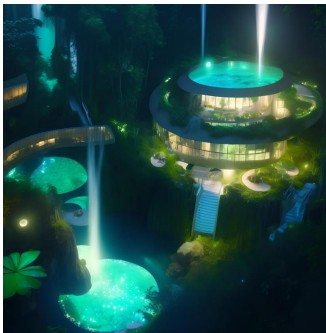

Utopian biophilic eco mansion design in a magical enchanted dream lush pine forest Trend on Artstation, Altered Carbon, waterfalls, streams, pools of golden glowing bioluminescent water, space ship take off, amazing night, photography, cinematic, aerial side groung view, Zoha Hadid.

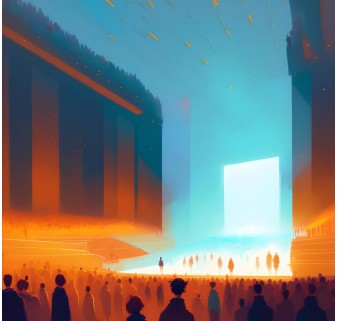

A crowd of people coloured and light in a theater, christopher balaskas, atey ghailan, mirror rooms, dynamic figure studies, misty, light cyan.

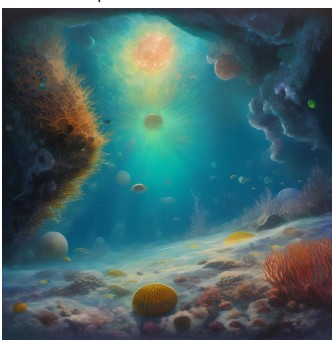

Realistic artistic astrobiological painting, a wide view of the sea floor under the waves of an alien ocean, where life has crawled out from the deep to cling to the sundappled seabed and begin the eternal dance of evolution into new and innovative forms.

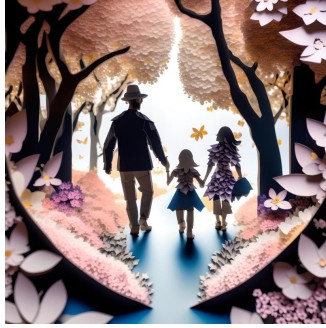

A man, a women and a child hand in hand, walking under blooming flower trees, miniature photo, overlook, paper kirigami craft, high detail.

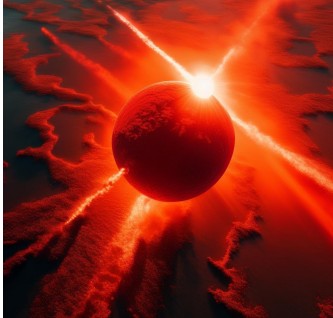

A red dwarf star showing a large blast of light, alien worlds, red, hyper-realistic water, deconstructive, angura kei, camera tossing, ansel adams, guy aroch, aerial view .

Figure 15: These examples show that RAPHAEL can generate artistic images with varying text prompts across various styles.

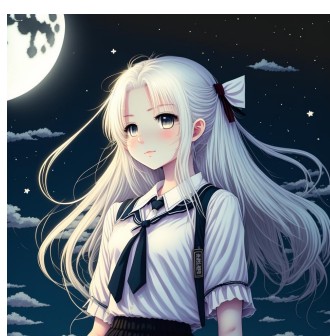

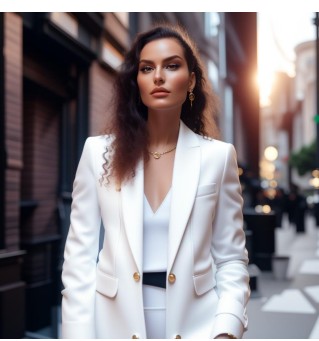

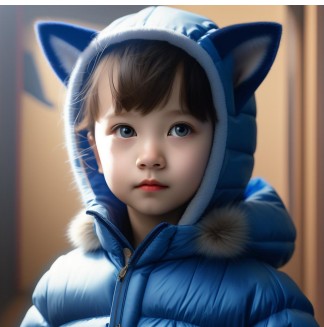

Celluloid style, Japanese manga style, cute girl in school uniform, long white hair, under the moon, night, high detail.

Photography, a woman wearing white blazers in a street, chic, copy space, fashion, low light, models, outfit, pose.

Photography, cute child, wearing a down jacket, cat ears, fleshy face, blue high details, detailed light and shadow.

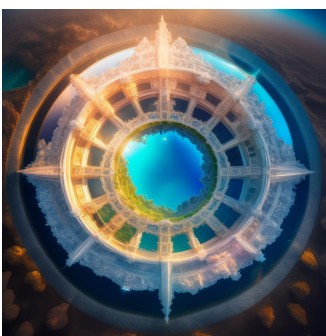

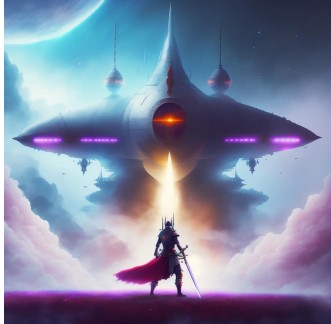

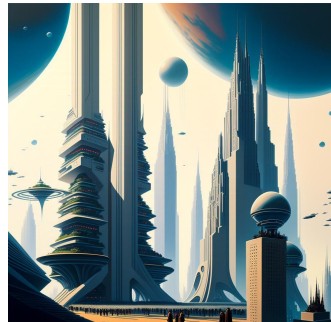

Bright scene, aerial view, ancient city, fantasy, gorgeous light, mirror reflection, high detail, wide angle lens.

Spaceship, a swordsman, magic, fantasy, fog, surreal, bright and brilliant light, gorgeous, glory, epic, high detail.

Minimalist sci-fi illustration, futuristic metropolis, tall skyscrapers, floating gardens, crowd, multi-level, detail.

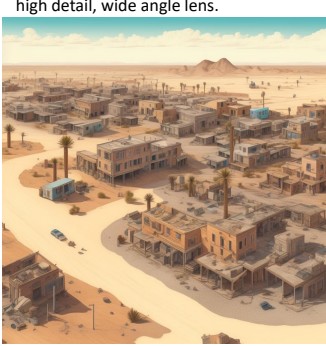

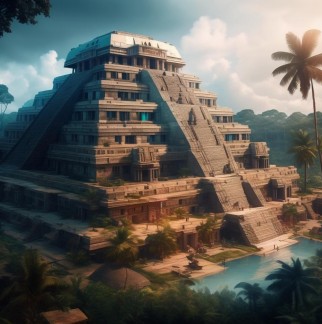

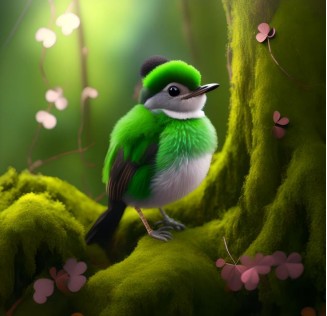

A picture on the browser of an abandoned desert town, collage - oriented, aerial view, ultra detail.

Mayan empire, use Mayan architecture, modern, technological cyberpunk style, photo quality.

A very cute little Shamrock bird in a mossy forest.

Figure 16: These examples show that RAPHAEL can generate artistic images with varying text prompts across various styles.

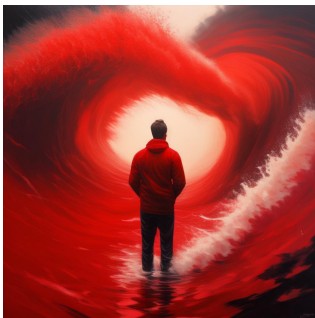

A painting depicting a red wave outside, trapped emotions depicted, full body, Jon Foster, depth, Dima Dmitriev, fisheye effects, Ray Collins.

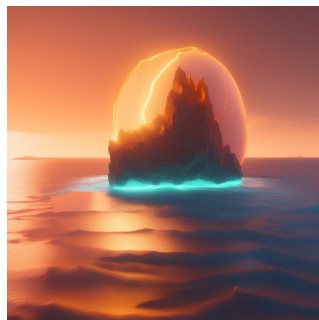

An enormous light next to the ocean, filip hodas, mike winkelmann, light amber and gold, glowing neon, mystical terrains, rectilinear forms, realistic lighting.

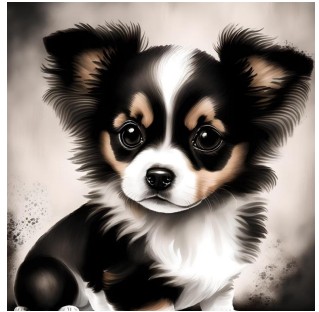

Papillon dog puppy in style of sumi ink painting, fantasy art, enigmatic, mysterious.

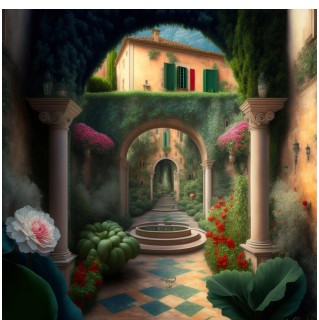

Italian secret garden, surrealism, high detail.

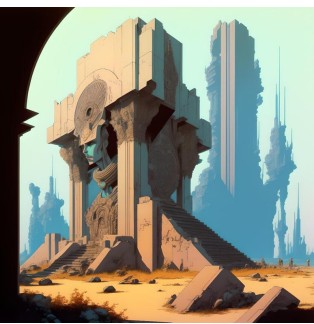

Ancient ruin Remnants of a religious temple, futuristic, machine-like lines, robotic motifs decorations, poster moebius style.

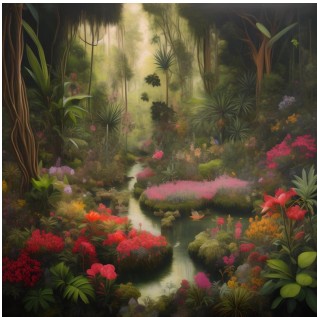

An intricate forest painting, full of exotic plants and flowers, Arianna Caroli.

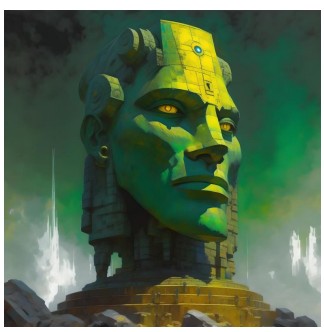

An ancient stone Colossus with eye, Stephan Martinière, dark yellow and light emerald, color zone painting, Denis Sarazhin, dark emerald and silver, robotic expressionism, high detail.

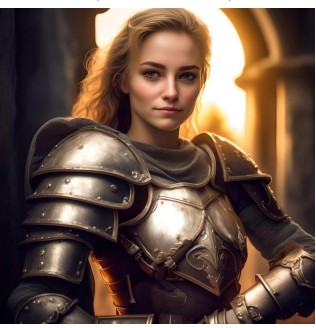

A knowing paladin, wry smile, feminine pose by the evening hour, foreshortening, gritty photographic, close enough to see the pores, determination emotion, photography, high detail.

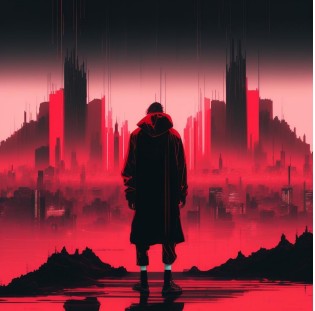

One man in the middle of a dark urban city, dark beige and red, futuristic landscapes, Schizowave, cryptidcore, radiant clusters, monumental scale, datamosh.

Figure 17: These examples show that RAPHAEL can generate artistic images with varying text prompts across various styles.

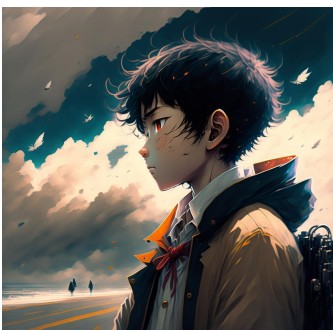

Celluloid style, Japanese manga style, flat-painted illustration, due teenager, big bag, short black hair fluttering in the wind, on the road by the sea, high detail.

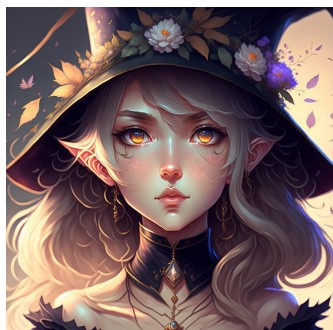

Impasto, Japanese and Korean manga characters, CG characters, CG avatars, magical girls, black magic hats, with flowers on the hats, elf ears, translucent, long gray hair, Slightly curly hair, trends on Artstation, high detail.

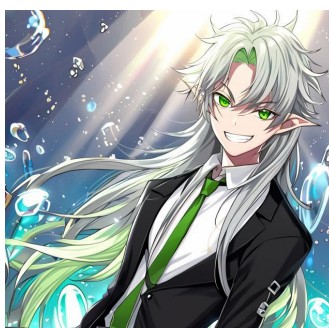

Flat painting, Japanese and Korean manga characters, male protagonist, elf boy, white and green hair, long hair, long ears, sea, bubbles, black suit and green tie, evil smile, green eyes, anime avatar, detailed light and shadow, high detail.

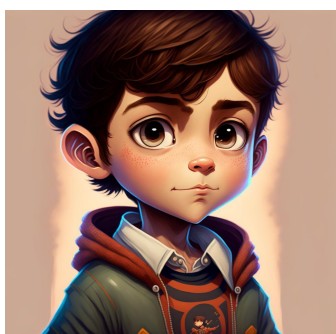

American comic character, flat painting, illustration character, avatar, flat painting illustration, short brown hair, curly hair, green clothes, ID photo pose, jane's style, trends on artstation , detailed light and shadow, high detail.

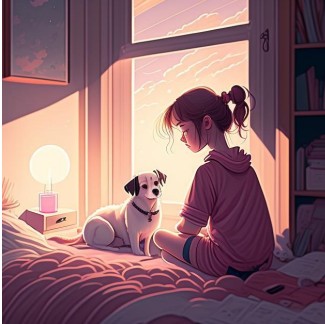

Flat drawing illustration, flat drawing, illustration, girl in pajamas sitting on the bed, a puppy, sunset, the light of the setting sun enters the house through the window, detailed light and shadow, high detail, 4k.

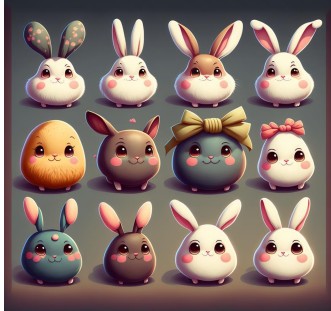

Cartoon character, mini character, figure, illustration, dango big family, bunny ears, different colors, character design, a set of avatars, cute, mini detail light and shadow, high detail.

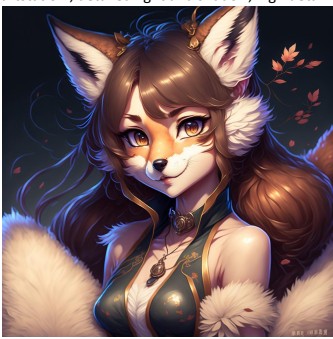

Furry character, character design, impasto, cute fox girl, long brown hair, curly long hair, fluttering in the wind, messy, big eyes, smile, petals fluttering in the wind, high detail, CG characters.

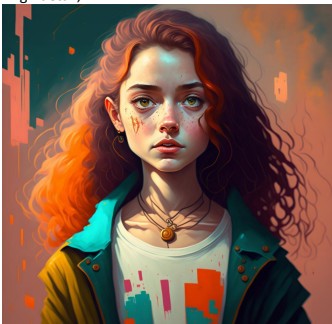

Impasto, avatar, illustration, girl with red hair, slightly curly hair, European and American, freckles, jane's style, trends on artstation, crazy colors, light and shadow contrast, high detail.

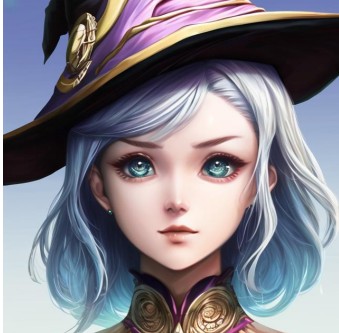

Impasto, Japanese and Korean manga characters, illustration characters, avatars, CG characters, CG avatars, magical girls, purple magic hats, short silver-white shoulder-length hair, curly hair. two-dimensional, detailed light and shadow, high detail.

Figure 18: These examples show that RAPHAEL can generate artistic images with varying text prompts across various styles.

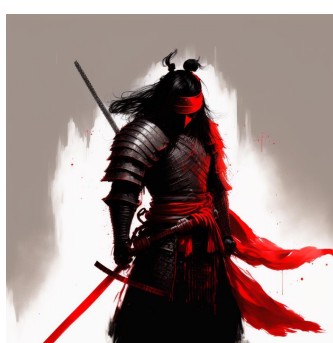

Character standing drawing, character design, impasto, samurai in black armor with long black hair, holding a long towel in his hand, blindfolded by a long cloth, CG game , game characters, high detail, color contrast.

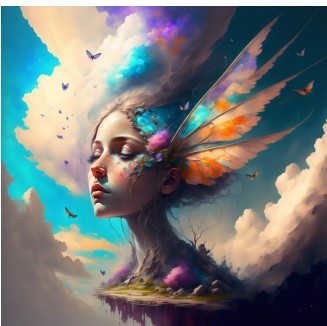

Impasto, avatar, illustration, color explosion, elf girl on the clouds, reflection, elf ears, colorful clouds, jane's style, trends on Artstation, light and shadow contrast, high detail.

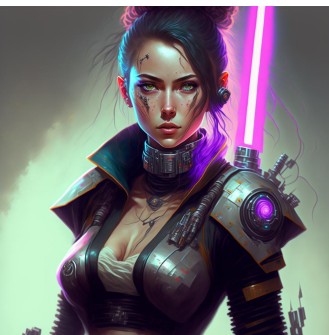

Impasto, illustration character, illustration poster, tech style, cyberpunk, woman with black long hair, hair up, black leather jacket, mechanical arm, tattoo on face, holding a purple lightsaber, war, trends on Artstation, high detail.

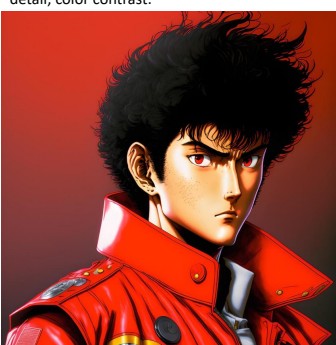

Spike Spiegel wearing Akira Shōtarō Kaneda's red jacket and outfit, Katsuhiro Otomo, cinematic, extremely detailed and complex, impressive, super resolution, megapixel.

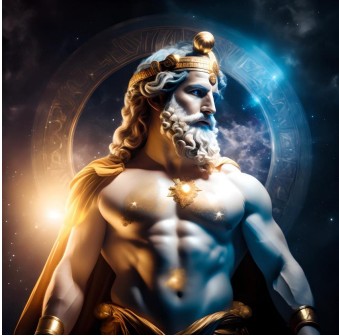

A Greek god, Erebus wearing ancient greek clothing, galaxy with a solar system as background, cinematic, soft studio lighting, backlighting, dark background.

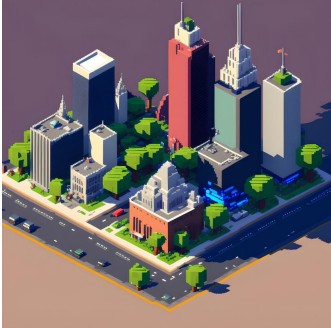

Pixel art, videogame city wallpaper, deskmat. io, low poly, large, wide engle , from the top.

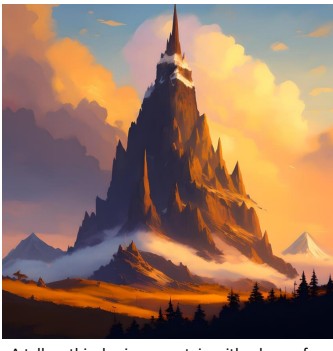

A tall mythical spire mountain with a base of clouds below during golden hour.

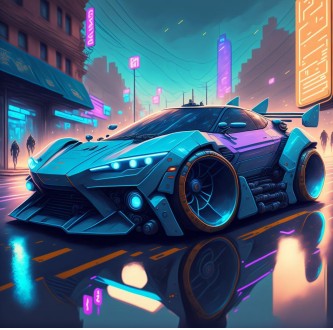

Atmosphere, illustration, mecha sports car, mecha era, running on the bustling street, blue, purple, cool.

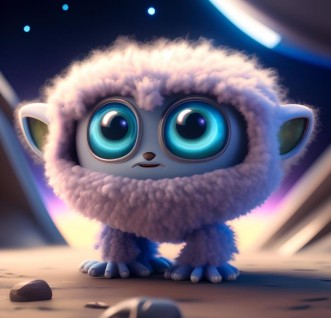

A cute fluffy sentient alien from planet Axor, in the andromeda galaxy, the alien have large innocent eyes and is digitigrade, high detail.

Figure 19: These examples show that RAPHAEL can generate artistic images with varying text prompts across various styles.

## 7.8 Extension to LoRA, ControlNet, and SR-GAN

We give the results of LoRA in Fig.20 and 21, ControlNet in Fig.22, and SR-GAN in Fig.23 and 24. The detailed settings are given in captions.

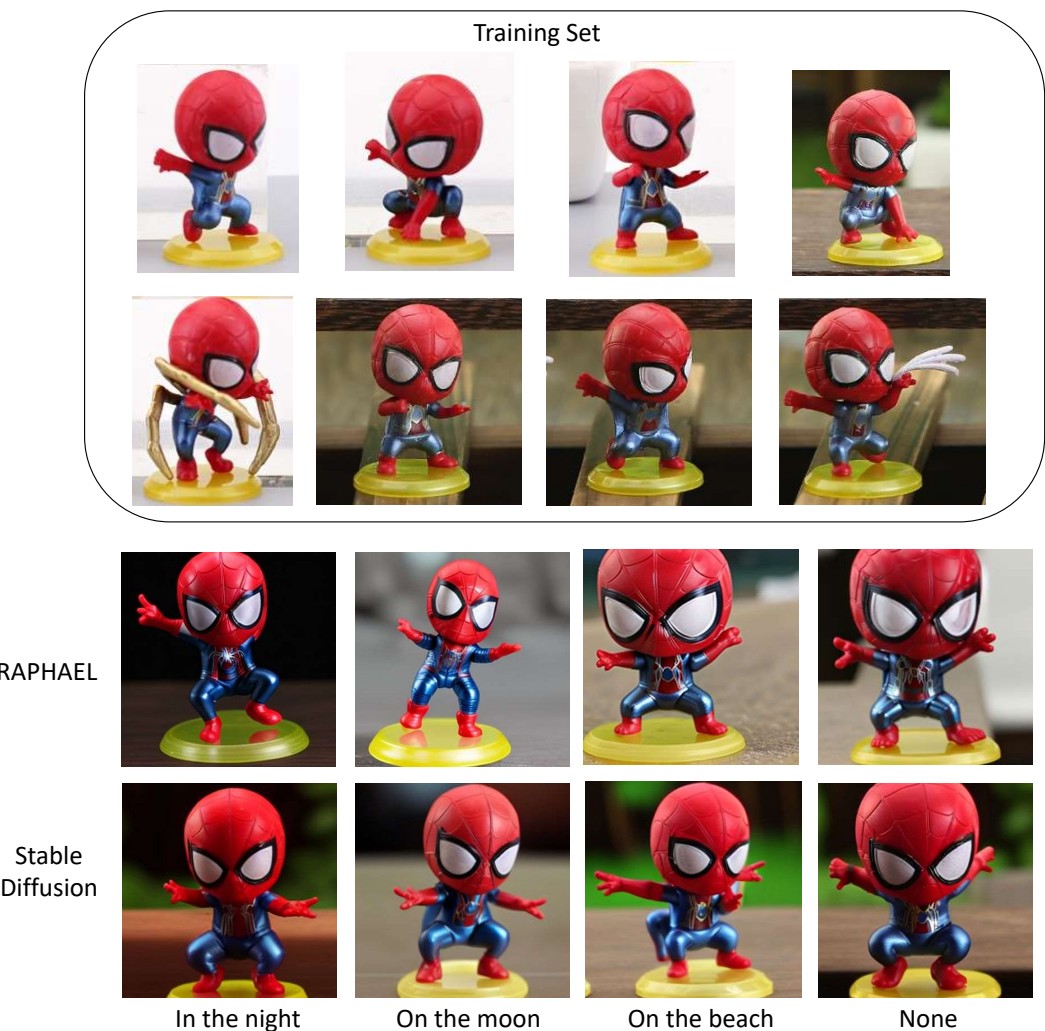

Figure 20: **Results with LoRA.** We use 28 images to finetune RAPHAEL and Stable Diffusion. The prompts are "A spider-man figurine, in the night/on the moon/on the beach/none", only RAPHAEL preserves the concepts in prompts while Stable Diffusion yields compromised results.

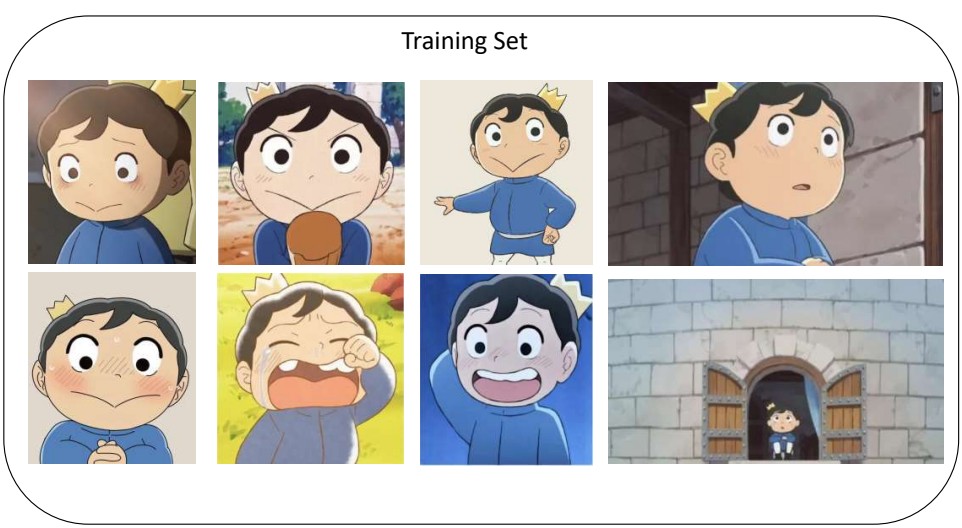

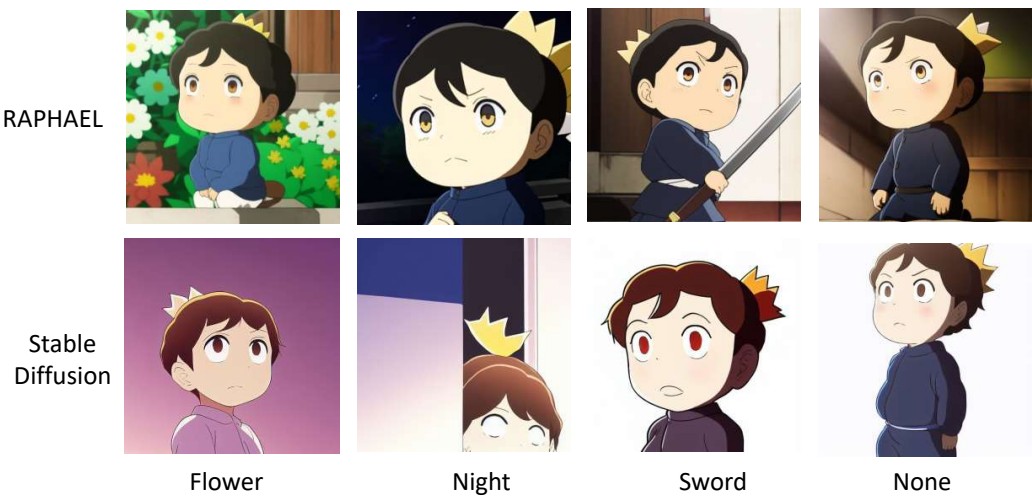

Figure 21: **Results with LoRA.** We use 32 images to finetune RAPHAEL and Stable Diffusion. The prompts are "A boy, flower/night/sword/none", only RAPHAEL preserves the concepts in prompts while Stable Diffusion yields compromised results.

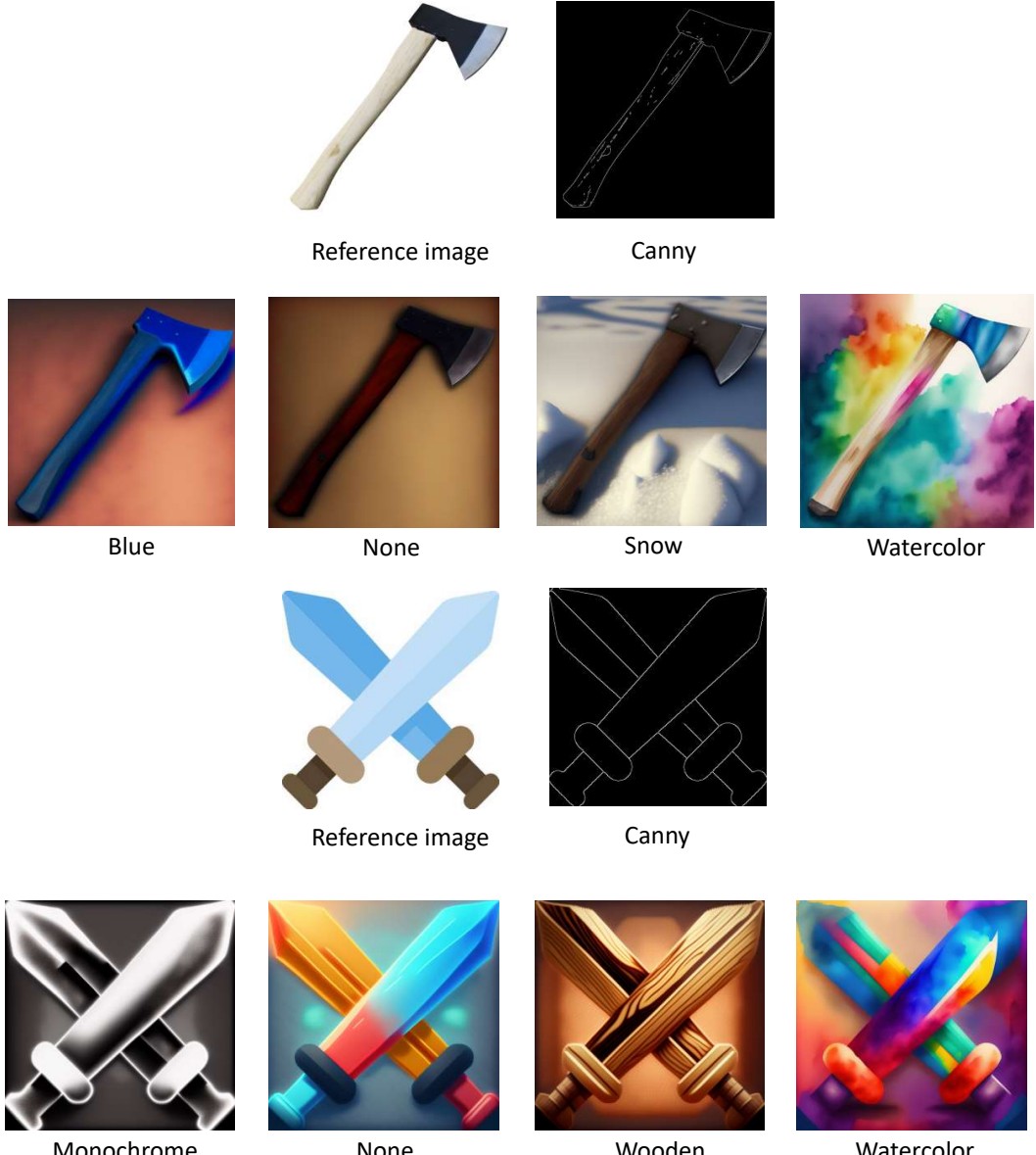

Figure 22: **Results with ControlNet.** We use the reference image to generate canny edges and adopt it as the extra constraint for RAPHAEL. The prompts for each group are "An ox, blue/none/snow/watercolor" and "Icon for game, fighting skill, monochrome/none/wooden/watercolor".

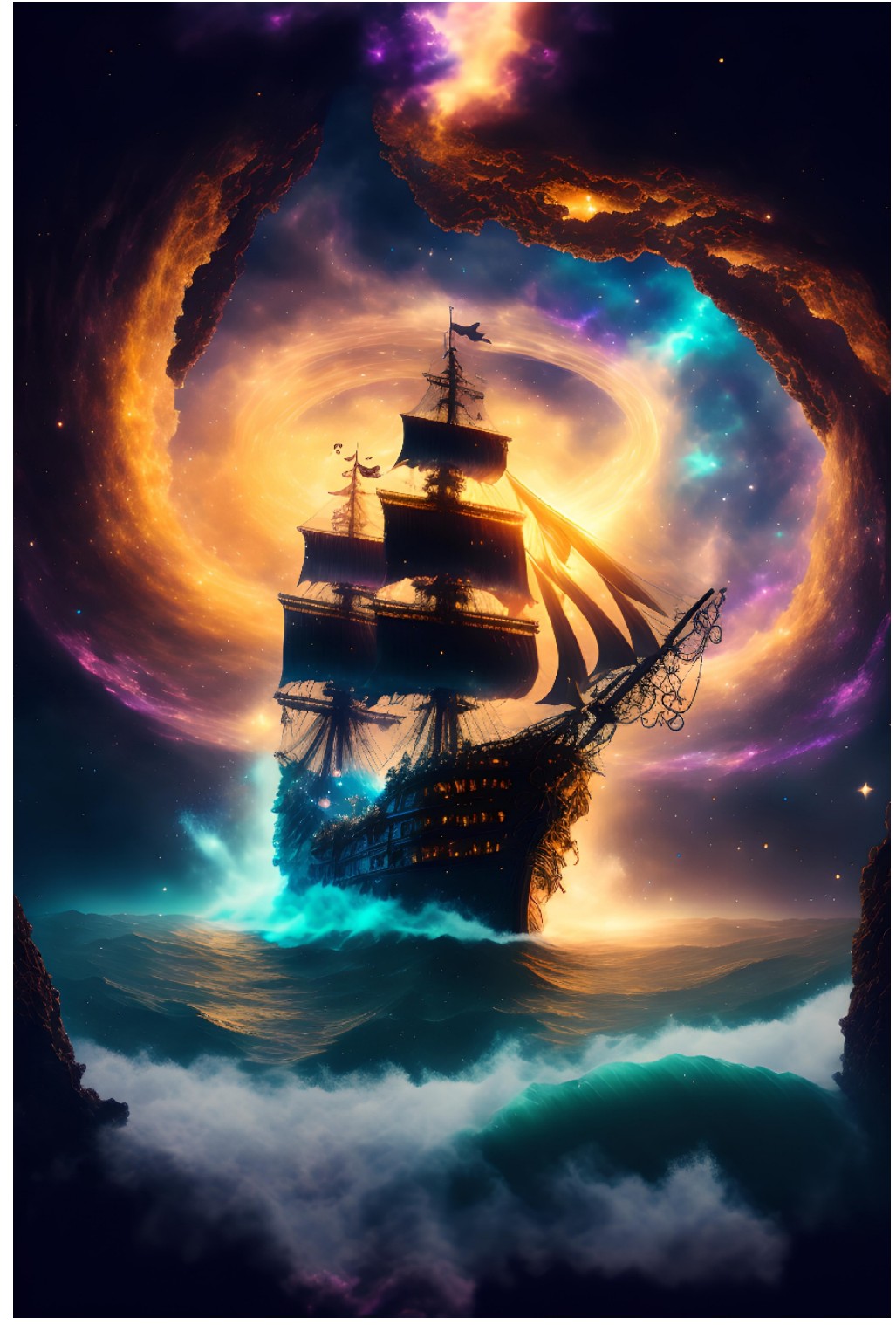

Pirate ship trapped in a cosmic maelstrom nebula, rendered in cosmic beach whirlpool engine, volumetric lighting, spectacular, ambient lights, light pollution, cinematic atmosphere, art nouveau style, illustration art artwork by SenseiJaye, intricate detail.

Figure 23: Result of 4096×6144 image. SR-GAN enhances the resolution of the image generated by RAPHAEL.

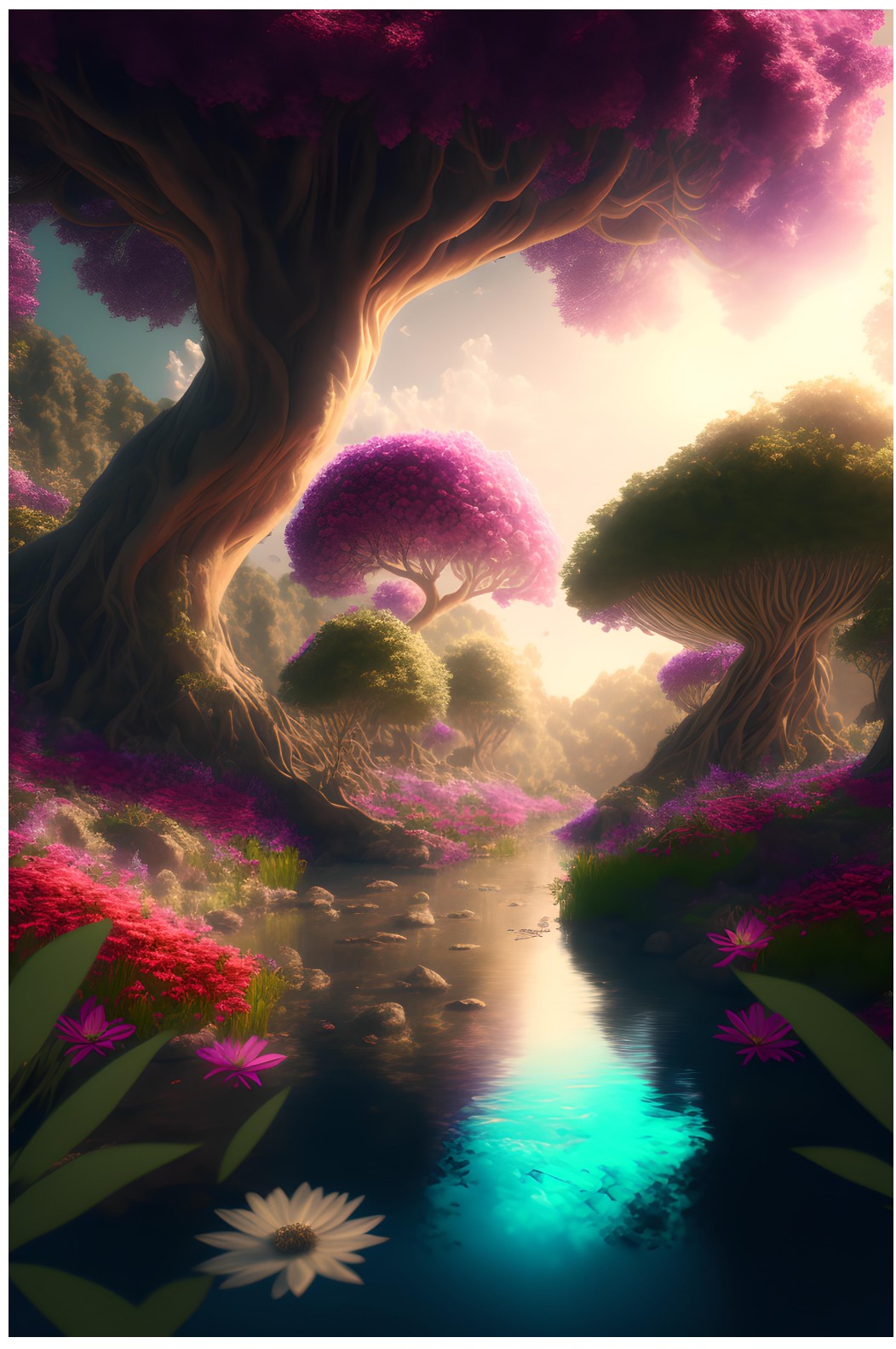

A sureal parallel world where mankind avoid extinction by preserving nature, epic trees, water streams, various flowers, intricate details, rich colors, rich vegetation, cinematic, symmetrical, beautiful lighting, V-Ray render, sun rays, magical lights, photography.

Figure 24: Result of 4096×6144 image. SR-GAN enhances the resolution of the image generated by RAPHAEL.