# OpenReview forum: "RAPHAEL: Text-to-Image Generation via Large Mixture of Diffusion Paths"
_NeurIPS.cc/2023/Conference — NeurIPS 2023 poster_

### Official Review · Reviewer_fnVX · 2023-06-27

**Soundness:** 3 good
**Presentation:** 3 good
**Contribution:** 2 fair
**Rating:** 5
**Confidence:** 4

**Summary:**

This paper presents a new large model RAPHEL (short for distinct image regions align with different text phases in attention learning ) for text-to-image generation. Technically, RAPHEL builds upon the LDM pipeline, with VAEs as image encoder-decoder, and then incorporates the MOE layers for spatial and temporal (in terms of diffusion steps) refinement in the diffusion generation process to improve the text and image fidelity. For the experiments and evaluations, the model is trained with LAION-5B and some confidential internal data, comparisons w/ other large models show better performance using the zero-shot FID on the COCO.

**Strengths:**

- The paper is well written with a clear structure and easy to follow.

- The model achieves sota FID and qualitative results compared to other strong and powerful similar scale models like Stable Diffusion, DeepFloyd and DALLE2.

- The high-level idea to incorporate MoEs to refine the spatial and temporal details in DPMs for text2image synthesis is intuitive and reasonable, with effectiveness proved in ablation studies.


**Weaknesses:**

- This is another large-scale model work that requires 1000 A100 GPUs with 2-month training on LAION-5B plus internal dataset, while the reviewer acknowledges the popularity of the topic and its superiority in performance, it is another work that can be hardly reproduced by most researchers in the field, especially w/ internal training data inaccessible for the community.

- In terms of the methodology design, while the idea to refine the generation process w/ MoEs is intuitive, the technical novelties are rather limited, and are largely limited to this specific text2img task, as there are several existing works w/ similar ideas [4,5].

- Some technical details remain rather coarse and unclear. See details in my questions.


**Questions:**

I have several questions regarding several technical aspects listed below:

- I am still confused on the working mechanism of time-MoE after reading the paper and appendix. For the time-MoE, it is a Time Gate Network at each diffusion step, located between the cross-attention and space MoE. The output of time-MoE is fed into the space-MoE, then what does this info depict for space-MoE at different diffusion steps?

- How does this work in inference, if the info from time-MoE does convey critical information in terms of steps, then how can the info be used in inference especially with the skipping sampling steps?

- What is the inference time cost using the proposed RAPHEL compared to other popular models?

- How does the internal data impact the final performance? Does the performance change evidently w/o the inaccessible internal dataset?


**Limitations:**

The paper discusses the limitations and potential negative impact on the risk of generating images with misleading and false information.

---

> ### Author Rebuttal · Authors · 2023-08-08
>
> Dear reviewer fnVX,
>
> Thanks for your comments. We will address your concerns below.
>
> **Q1: RAPHAEL uses internal datasets and many computing resources**
>
> We argue that this is not a weakness for rating specific to the text-to-image diffusion model community. Numerous academic papers accepted by top conferences/journels have internal datasets. We provide a table on text-to-image diffusion models here:
>
> | Model          | Venue                        | Internal data | GPUs/TPUs |
> |----------------|------------------------------|---------------|-----------|
> | Ours           | N/A                          | Yes           | 1k  A100s      |
> | Imagen         | NeurIPS'22 Outstanding Paper | Yes           | 512    TPUs   |
> | ERNIE-ViLG 2.0 | CVPR'23                      | Yes           | 320  A100s     |
>
> **Notably, Imagen even received the outstanding paper award at NeurIPS last year.** We also understand ERNIE-ViLG 2.0 uses fewer A100s than us, but they continue to update the model from last year's September and we use their latest API to make comparisons. So it's difficult to compare the GPU hours.
>
> Moreover, we intend to address this issue by releasing an API to make the model more accessible to the public. We firmly believe that RAPHAEL will contribute significantly to the advancement of text-to-image generation in the research community.
>
> Other than the diffusion models mentioned above, **other text-to-image models also use internal datasets, such as MUSE (ICML'23), GigaGAN (CVPR'23), Parti (TMLR'22)**.
>
> **Q2:The novelty is limited.**
>
> As highlighted by reviewers PszJ, **"The space-MoE technique, specifically in the context of text-to-image generation, appears to be new and well-motivated, leading to a meaningful performance boost." and "The main technical contribution is well-motivated and novel."** Reviewer NLKN also claims "There are **two novel ideas** explored in the work: using spatial MoEs (for attended features) and using edge supervision for attention weights. Both ideas make sense and should be easily extendable to other setups. " Both reviewers acknowledge that RAPHAEL presents novelty.
>
> The space-moe and edge-supervised learning in RAPHAEL are our original contributions, which conduct region-level refinement during the denoising process and significantly improve image quality. We have also implemented a gating function to enhance the performance of time-moe, as noted by reviewer PszJ: "I particularly appreciate the gating mechanisms, where, in the case of time-MoE, they automatically assign different timesteps to various time experts. Previous work manually assigned experts to different diffusion time intervals (e.g., eDiff-I)." Our experts are assembled on-the-fly during inference, which is more flexible and parameter-efficient.
>
> **Q3: Working mechanism of time-MoE**
>
> The space-moe and time-moe are disjoint. We experimented with different placements of space-moe and time-moe and obtained similar performances.
>
> The feature processed by time-moe is passed on to space-moe. This feature serves as a better representation of latent features than before and is utilized in space-moe. The gating function takes text tokens as input to determine which experts should process the features, as output by the output of time-moe. Notably, the space-moe does not have any temporal dimension; it leverages the improved feature representation from time-moe.
>
> **Q4: Time-moe in inference**
>
> During the training process, the gating function is trained to automatically select the appropriate time expert. Once convergence is achieved, for example, at time step 1, expert 1 assists the UNet in fitting the score function. At time step 500, expert 2 is assigned to fit the score function. Therefore, during the sampling process, each time step is associated with a specific time expert to fit the score function. Although skipping of some time steps may occur during sampling when using DDIM or DPM solvers, the score functions for the selected time steps are accurately fitted with the assistance of these experts.
>
> **Q5: Inference cost**
>
> We provide an analysis in Section 4.2, which shows that the inclusion of space-moe results in an additional 24% overhead. This is faster than models with cascaded designs, such as Imagen and DeepFloyd. Furthermore, time-moe and edge-supervised learning do not introduce any extra inference cost.
>
> We can also compare RAPHAEL with other popular models. All the models provided here are highly optimized, so we will choose our optimized version to compare (a bit faster than the results in Fig 6c because of the update of our infrastructure). We also use this same environment to conduct a fair comparison between RAPHAEL and Stable Diffusion XL. **Please refer to the pdf file in the general response for the results.** So we think the inference cost is not the bottleneck for RAPHAEL.
>
> **Q6: The impact of internal datasets**
>
> We incorporated internal datasets to improve the aesthetics of the generated images, a practice common among prestigious text-to-image models, including Imagen, Stable Diffusion XL, DeepFloyd, MUSE, Parti, ERNIE-ViLG 2.0, DALL-E 2, eDiff-I, etc.
>
> The internal data primarily impacts the aesthetics of the generated images. Due to the limited time of the rebuttal period, we resume the checkpoint of RAPHAEL trained for two months, and continue to train it with LAION-5B for ~7 days instead of training from scratch. It's also reasonable because of the catastrophic forgetting properties of deep neural networks. Based on this model, we conduct a human evaluation using ViLG-300. The results indicate that most people (72.6%) prefer the model trained on internal datasets in terms of image quality. This preference is attributed to the poor image quality of LAION. However, we don't observe significant difference in image-text alignment.
>
> Additionally, we measure the FID based on this model, resulting in 6.79. So the internal data doesn't have much influence on the coco-30k FID.

---

> > ### Comment · Reviewer_fnVX · 2023-08-15
> >
> > I would like to first thank the authors for the rebuttal and for carefully responding to my questions and concerns.
> >
> > I have read the rebuttal and (have always) acknowledged that the RAPHAEL is a technically solid work with its valuable contributions to the community. However, I also do not think the fact that other similar works, such as Imagen, have been previously awarded with the prize, or other large generative models have also used internal datasets is the expected answer to my raised concern (to be honest, it is never a question or concern specifically against RAPHAEL but a general research question/concern), in terms of reproducibility, model accessibility or performance. Personally, I would be more appreciative of works that actually improve our understanding of the model itself and can be beneficial and insightful for other practical and widely accessible applications.
> >
> > Despite the above, I think my questions regarding the technical details have been answered, and thus raised my score after rebuttal.

---

> > > ### Author Response · Authors · 2023-08-15
> > > **Thanks for your feedback**
> > >
> > > We deeply appreciate your valuable feedback. And we will incorporate the technical details uncovered during the rebuttal into the final version if the paper is accepted

---

### Official Review · Reviewer_NLKN · 2023-07-05

**Soundness:** 2 fair
**Presentation:** 2 fair
**Contribution:** 2 fair
**Rating:** 7
**Confidence:** 4

**Summary:**

The paper trains a large-scale latent diffusion model for image synthesis. It trains on a mix of LAION-5B (post-processed + filtered by aesthetic score) and in-house data. It proposes two novel technical contributions: 1) using a "spatial" mixture-of-experts (MoE) where an expert is predicted from each text token and processes the attended features; and 2) supervising attention weights via edge maps. Interestingly (and differently from most of the prior works), it also uses multi-scale training. Qualitatively, the generated images look substantially better than the ones from the baselines. It achieves SOTA FID on COCO, which is the main metric/benchmark for text-to-image generators.

**Strengths:**

- The method achieves the very best known results for large-scale text-to-image synthesis (among those models which could be rigorously benchmarked against).
- There are two novel ideas explored in the work: using spatial MoEs (for attended features) and using edge supervision for attention weights. Both ideas make sense and should be easily extendable to other setups. For spatial MoEs, there are also test-time visualizations provided which helps in understanding their influence.
- The comparison to other methods is careful and thorough: a lot of non-cherry-picked qualitative results provided; human studies are performed, quantitative metrics are reported.



**Weaknesses:**

I have two big concerns: 1) spatial MoE and edges supervision are not properly ablated; and 2) the paper does not contain enough technical details to be reproduced. I will elaborate on them below:

1. Improper ablations. After spending ~5 hours on reading the paper, it's still not clear to me where exactly the SotA FID score on COCO is coming from — architecture, data, or optimization, since they are all intermixed in the final model. Fig 6 denotes several ablation experiments, but it's not specified anywhere how were they trained. Does each run in Figure 6 was trained for 2 months on 1,000 GPUs as well? On the same dataset? Such lack of details about ablations makes them impossible to understand and analyze. For Figure 6c, what the FID scores for the line "Computational complexity" denote? This is confusing.

2. Lack of details. The current manuscript is something between an academic paper and a technical report. Here are some (of many) missing details:
  - How exactly does your U-net and VAE look like? Are they equivalent to LDM ones, but with larger channel sizes? Or there are other modifications (apart from spatial/time MoEs)?
  - How many images are in your in-house dataset, what are their resolutions and how it was collected?
  - How exactly was multi-scale training implemented? What are the resolution distributions in your final dataset? Do I get it right that different batches on different GPUs have different amount of images in them (since the resolutions are different)? Do you allocate the same amount of GPUs per bucket? Do I get it right that your VAE is multi-scale, while the diffusion model is not? Or vice-versa?
  - How random noise \epsilon is sampled for expert routing (L122)?
  - What is the motivation of using focal loss for edge prediction instead of other loss types? Using focal loss here is quite non-intuitive to me. Did you try ablating it?

There are also several smaller (but still reasonable) concerns:
- It's not clear whether spatial experts reflect any text semantics. Judging by Figure 4, the expert assignments are completely random. I would expect to see that visually similar concepts ("tiger"/"cat"/"leopard"; "dog"/"wolf"/"fox"; "tv"/"monitor") would get assigned to the same expert. Could you please provide any support or refutation to such intuition (e.g., buy checking the clusters)?
- The writing quality could be improved. There are many variables introduced, and it would ease reading if they would be described in text, i.e., instead of writing "we set \lambda to X", one should write "we set learning rate \lambda to X". Otherwise, a reader needs to jump back and forth trying to recall the variable meaning.
- There is a quite confusing notation clash:
  - \epsilon denotes random noise in diffusion and random noise (L74) in experts routing (L122)
  - \alpha denotes variance schedule (L74), focal loss hyperparameter (Figure 6a), routing multiplier (L129)
- Limitations are not properly discussed (see the "Limitations" form below)
- GigaGAN is a missed reference since they also use MoEs, routed by text tokens.
- It's not clear why SR-GAN is included into the exposition since there is nothing special about it, and SR-GAN can be combined with any other image generator. Does it work better for RAPHAEL than for other image generators? If so, then it's interesting. But if not — then it's not clear why claiming that RAPHAEL can generate 4096 x 6144 images when combined with it. Following such arguments, what prevents one saying that StyleGAN can generate 100,000 x 100,000 images when combined with tailor-mode bilinear interpolation?

Typos and minor comments:
- L114: "mean of all experts" => "weighted average of all experts"?

I look forward to discussing my concerns with the authors and fellow reviewers and improving my rating.

**Questions:**

I raised several concerns in the "Weaknesses" section and would be grateful to hearing the author's opinion on them. My main concerns are the lack of details about the method and experiments.

**Limitations:**

There is a brief limitations discussion on potential negative societal impact. In this regard, I wouldn't demand more discussion from the authors since a potential misuse of powerful image generators is a well-known issue to the community and should be discussed at a "higher" level, rather than in this particular work.

However, it would be good to see the discussion of other potential limitations of the work, for example, what are the disadvantages of binarizing the attention maps? whether it is possible to ablate the model properly — at least via convergence plots for partial runs (I understand that training a full model for each ablation is infeasible)? what could be a problem with edge maps supervision of the attention maps (i.e., I guess there should be failure cases in edge map detection)? And so on.

---

> ### Author Rebuttal · Authors · 2023-08-08
>
> Dear Reviewer NLKN,
>
> Thanks for giving so many constructive suggestions for our paper, I will clarify the settings.
>
> **Q1: The setting of ablation study**
>
> We conducted an ablation study in the following manner:
>
> For Fig.6a, we resume the final model trained for two months, and train each point in Fig.6a for another 100M samples to ensure convergence, using the same dataset (LAION and internal datasets) and seed.
>
> For Fig.6b, we resume the final model trained for two months and then individually deleted the space-moe, time-moe, and edge-supervised learning modules, resulting in three different models. Next, we continued to train each of these models without the respective module with 100M training samples **to ensure their convergence**, using the same dataset (LAION and internal datasets) and seed. **The implementations of space-moe and time-moe are both with residual connection, so the deletion operation is reasonable.** We measure the FID curves for these three models, and the results are presented in Fig. 6b.
>
> For Fig. 6c, the red and blue lines represent the FID-expert curves (left axis) for space-moe and time-moe, respectively, showing that FID decreases with an increased number of space experts and time experts. We conduct this ablation study following the pipeline of Fig.6b.  We resume the final model trained for two months and then individually deleted the space-moe or time-moe. We add new space-moe or time-moe modules according to the number of experts needed and train each setting for 100M training samples.
>
> The green line in Fig. 6c represents the inference speed (DDIM steps/s, right axis) with an increased number of space experts. We provide an alternative way to speed up space-moe in our cluster. The definition of space-moe is as follows: $\frac{1}{n_y}\sum_{i=1}^{n_y} e_{\operatorname{route}(y_i)}(h'(x_t) \circ \widehat{\mathbf{M}}_i )$. Each token's corresponding feature will be routed to different experts naively, which is achieved through a "for" loop and cannot be optimized in our hardware setup. We propose an implementation approach as follows to address this:
>
> 1.We obtain a list $[a_1, a_2,..., a_k]$ = $[e_1(h'(x_t)), e_2(h'(x_t)),..., e_k(h'(x_t))]$.
> 2. The output of space-moe can be calculated as: $\frac{1}{n_y}\sum_{i=1}^{n_y} a_{\operatorname{route}(y_i)}\circ \widehat{\mathbf{M}}_i$.
>
> The above implementation is always faster. We also ablated these two implementations, each trained with ~120M samples, and find that they exhibit similar performances in human evaluation and similar diffusion paths. The inference cost compared with other popular text-to-image diffusion models can be found in the pdf file of the general response.
>
> **Q2: * How exactly does the U-net and VAE look like**
>
> Please refer to the Q3 of our global response.
>
> **Q3: In-house dataset**
>
> Please refer to the Q1 of our global response.
>
> **Q4: Implementation of multi-scale training**
>
> Please refer to the Q2 of our global response for the detailed implementation.
>
> **Q5: Choice of \epsilon.**
>
> A small number 1e-6.
>
> **Q6: Choice of Focal Loss**
>
> Yes, we have explored other loss types and conducted ablation studies to address this. The most intuitive loss is cross-entropy loss. However, we encounter imbalanced edge-maps and background, where the prediction module tends to classify all pixels as background, negatively impacting the cross-attention maps. To overcome this, we decide to adopt Focal Loss, which effectively handles the imbalance issue.
>
> **Q7: Semantics of space experts.**
>
> We observe some patterns, similar concepts have similar diffusion paths (each concept generates 100 paths with our template). For example, smooth/glossy, minimal/minimalist, dreamy/dreamlike, happy/joyful, sad/gloomy, brave/courageous, tired/exhausted, bright/luminous, honest/sincere, puzzled/confused, brilliant/shining, grateful/thankful, harsh/severe, enormous/huge, humble/modest. We find the diffusion paths generated by each pair always share at least 11 experts out of 16 blocks.
>
> For the diffusion paths given by visually similar but **different** pairs, such as tiger/cat/leopard (300 paths); dog/wolf/fox (300 paths); tv/monitor (200 paths), they also always share a relatively small number of experts (less than 7). We guess it is because they don't have similar semantics.
>
> **Q8: Writing quality, notations, references and SR-GAN**
>
> Thanks for pointing out these issues. We will polish it in the future since we can't edit the paper in the openreview now.
> We will also modify the claim about SR-GAN.
> And sorry for missing this excellent paper GigaGAN. We will add it once we can edit the paper.
>
> **Q9: Limitations**
>
> Indeed, we acknowledge several limitations in our paper that require attention. One limitation is the direct binarization of the attention map, which may result in the loss of some information. An adaptive module should be proposed to address this issue effectively. Additionally, the performance may be affected by failure cases of the edge detector, leading to potential degradation. We plan to explore solutions for these limitations in our future work.
>
> Another limitation pertains to the design of our ablation study. Conducting each full setting in the ablation study would be prohibitively expensive and time-consuming. As a result, we opt to run partial settings for the ablation study, which still takes 1.5 months to complete. We recognize the need for a better and more efficient approach to conducting ablation studies of foundation models in the research community.
>
> During the 7-day rebuttal period, it is almost impossible to run so many experiments for re-plotting the convergence curves for Fig. 6. However, based on our experience, we observe that edge-supervised learning can converge in less than three days. On the other hand, space-moe and time-moe converge much slower, requiring at least two weeks, given the setting of 6 space experts and 4 time experts.

---

> > ### Comment · Reviewer_NLKN · 2023-08-13
> > **Follow-up questions**
> >
> > Thank you for your response, it helped me to understand your work much better and resolved several concerns.
> >
> > I would like to clarify a couple of more questions if that's possible:
> > 1. Do I get it right that for your multi-scale training, the diffusion UNet model also operates on inputs/outputs of different resolutions? E.g. an image of shape [3, 448, 832] is encoded into the latent code of shape [12, 56, 104]? Does this also mean that there is an attention layer somewhere inside the UNet running on a 7x13 resolution? Or you use different downsampling factors inside the UNet?
> > 2. What does "Use checkpoint" mean in Table 2 in the appendix?
> > 3. How many overall steps/epochs do you do over the course of your 2 months training?
> >
> > For my previously raised concerns:
> > - About Q1 (ablations). Honestly, the provided ablations are quite unusual, since the performance might drop after removing the components/objectives simply because it's a too drastic change in the architecture (even with the presence of residual connections). For Figure 6a, do I get it right that you trained the model with $\alpha=0.2$, $T_c=500$, and then, after 2 months of training, fine-tuned for other hyperparameter values? If so, how did you know that $\alpha=0.2$, $T_c=500$ were optimal?
> > - About Q7. Is it possible to attach any plots to the rebuttal or update the previously attached PDF? I would be curious to see the diffusion paths for those kinds of prompts.

---

> > > ### Author Response · Authors · 2023-08-14
> > > **Response to follow-up questions**
> > >
> > > Thanks for your follow-up questions. I'm delighted to address your concerns.
> > >
> > > **Q1: Multi-scale Training**
> > >
> > > Yes, the diffusion UNet model also operates on inputs/outputs of different resolutions.
> > >
> > > Yes, there is an attention layer somewhere inside the UNet running on a 7x13 resolution. Consequently, this necessitates the scales of buckets to be divisible by 64. Notably, my experience has shown that cropping images to a fixed scale (e.g. 640) can **destroy** the performances of text-to-image diffusion models. Therefore, the importance of adopting a multi-scale training approach becomes more pronounced.
> > >
> > > Based on my experience, fine-tuning a model trained at a fixed scale (e.g. 640 x 640) to a multi-scale version takes 5 days.
> > >
> > > **Q2: Use of Checkpointing**
> > >
> > > The term "use_checkpoint" originates from the configurations within the stable diffusion (unet_config). Pytorch Checkpointing is employed to conserve memory and mitigate CUDA out-of-memory errors.
> > >
> > > **Q3: Iterations**
> > >
> > > Our model has undergone training for approximately 1.02 million iterations, equivalent to roughly 2.9 epochs.
> > >
> > > **Q4: Ablation Study**
> > >
> > > **Experimental Setup**
> > >
> > > Firstly, the objective of the ablation study is to validate the efficacy of each module outlined in our paper. The deletion operation works very well in our experiments. Fig. 6c provides further affirmation – **by resuming training from the same initial point and progressively increasing the number of experts, a consistent reduction in FID is observed. This unequivocally confirms the efficacy of both space-moe and time-moe.**
> > >
> > > The deletion of the edge-supervised learning module doesn't significantly alter the architecture, and this module exhibits relatively rapid convergence.
> > >
> > > Secondly, we believe this to be the **optimal** approach for ablating RAPHAEL. This is particularly because our model is designed for a **text-to-image** and **large-scale** image generator to measure the **zero-shot** FID. Running these partial settings of ablation study has also entailed a substantial investment of 1.5 months and millions of dollars.
> > >
> > > **Tuning $\alpha$ and $T_c$**
> > >
> > > Regarding these two hyperparameters, we adopt a proxy-based strategy to identify the optimal values before commencing full-scale training.
> > >
> > > As both space-moe and edge-supervised learning align with stable diffusion, we incorporate them into stable diffusion v1.4. This involves training each configuration over a 14-day period, and subsequently searching the hyperparameters based on the evaluation outcomes. It's important to note that this isn't an ablation pertaining to RAPHAEL but rather an investigation based on stable diffusion. As such, the inclusion of this aspect in our paper is unfeasible. However, if necessary, we can upload this part to the appendix for the camera-ready version if the paper will be accepted since we can't edit it now.
> > >
> > > This practice is widely adopted within the foundation model community prior to embarking on full-scale training.
> > >
> > > **Q5: Diffusion Path**
> > >
> > > I have lost the permissions to edit any contents of my paper, PDF files, and even the preceding rebuttal during this author-reviewer discussion phase.
> > >
> > > I will definitely add these visualization results to our camera-ready version to facilitate the community and enhance the accessibility of our work if the paper will be accepted.
> > >
> > >
> > > Please let me know if you still have more concerns, and I'm very happy to discuss them with you.

---

> > > > ### Comment · Reviewer_NLKN · 2023-08-14
> > > > **Thank you for additional clarifications**
> > > >
> > > > I am grateful for receiving so many clarifications. It greatly resolved my concern about the paper missing important details. Also, they have
> > > >
> > > > > we believe this to be the optimal approach for ablating RAPHAEL
> > > >
> > > > I do not agree with this statement. You could have just trained a smaller model on something like MS-COCO for a 1-2 weeks on a couple of nodes. If the proposed technique is general enough, then it would lead to a superior quality in this setup as well. Or you could have trained a much smaller model on your internal dataset.
> > > >
> > > > > by resuming training from the same initial point and progressively increasing the number of experts, a consistent reduction in FID is observed
> > > >
> > > > That's a strong and valuable observation, I apologize for missing this out.
> > > >
> > > > > As both space-moe and edge-supervised learning align with stable diffusion, we incorporate them into stable diffusion v1.4. This involves training each configuration over a 14-day period, and subsequently searching the hyperparameters based on the evaluation outcomes. It's important to note that this isn't an ablation pertaining to RAPHAEL but rather an investigation based on stable diffusion. As such, the inclusion of this aspect in our paper is unfeasible.
> > > >
> > > > I do not see why the inclusion of this is infeasible. Showing that the proposed technique works for Stable Diffusion would be valuable for a reader. I think that the paper would benefit from incorporating such exploration on top of SD into the appendix, since it would show that space MoE works in a slightly more general setting, rather than for RAPHAEL only.
> > > >
> > > > Btw, is it possible to re-export the plots as PDF rather than PNG in the paper? The text on it (e.g., on Figure 5) is not selectable/searchable.
> > > >
> > > > --------------------------------------------------------------------------
> > > >
> > > > Overall, I believe that if the final version will incorporate all the details uncovered in the rebuttal and also additional visualizations for diffusion paths (as discussed in the previous response), it will be an interesting read for the community. Especially given that it even outperforms Midjourney (though it was not confirmed with human evaluation, only with qualitative evaluation on the selected prompts?). This is why I decided to increase my rating to "Accept".

---

> > > > > ### Author Response · Authors · 2023-08-14
> > > > > **Thanks for your comments and suggestions**
> > > > >
> > > > > We sincerely thank the reviewer for the constructive feedback and the kind support of this work! I will incorporate these details into the final version if the paper is accepted.

---

### Official Review · Reviewer_8jJJ · 2023-07-06

**Soundness:** 3 good
**Presentation:** 3 good
**Contribution:** 2 fair
**Rating:** 6
**Confidence:** 4

**Summary:**

The paper proposed RAPHAEL, a text-to-image diffusion model. The model adopt MoE layers, including space-MoE and time-MoE layers.  In addition, edge-supervised learning is proposed to enhance performance. RAPHAEL establishes a new state-of-the-art with a zero-shot FID-30k score of 6.61 on the COCO dataset, and surpasses its counterparts in human evaluation on the ViLG-300 benchmark.

**Strengths:**

1. The authors promised to release a programming API for RAPHAEL to the public.
2. Explore spatial- and time-moe, and perform some abaltion study.
3. visualize spatial- and time-moe in the appendix.
4. achieved sota zero-shot FID score on coco.

**Weaknesses:**

As a paper focusing on pretraining, more clarification of experiment details is needed. Including:

a. data. The paper mentioned that "The training dataset consists of LAION-5B and a few internal data.". How many internal data is used? Its category distribution and collecting sources?

b. model structure and hyperparameters. including VAE, and each stage of diffusion model.

**Questions:**

1. Is the space-moe performed only on text-image cross attention? How about applying moe to self-attention within image?
2. In abstract the authors mentioned that "RAPHAEL exhibits superior performance in switching images across diverse styles" (L13). Is it benefit from spatial-moe? if so, is there any pattern in moe paths regarding different styles?

**Limitations:**

see weakness and questions.

---

> ### Author Rebuttal · Authors · 2023-08-08
>
> Dear Reviewer 8jJJ,
>
> Thanks for appreciating our work and your advice. We will address your concerns below.
>
> **Q1: The paper mentioned that "The training dataset consists of LAION-5B and a few internal data.". How many internal data is used? Its category distribution and collecting sources?**
>
> The data consists of approximately 440 million entries filtered from LAION-5B and approximately 290 million entries from internal datasets, which is less than what Imagen's dataset contains. These datasets possess special characteristics, notably high-quality and aesthetics.
>
> To collect our internal datasets, we follow the methodology of DALL-E [1]. We curate a dataset on a scale similar to JFT-300M by sourcing images from the Internet. We remove instances with aspect ratios outside the range of [1/2,2], and we follow Stable Diffusion v1.4 to filter out images with low aesthetics scores. For captioning these images, we utilized BLIP-2. The main reason for constructing such an internal dataset with **high aesthetics** is to compensate for the poor quality of LAION. We don't add additional limits on categories but want to collect datasets with high aesthetics.
>
> Regarding the resolution distribution, adopting the buckets mentioned in the global response Q2, it is as follows: [52,24,161,470,52,81,56,70,34]
>
> Furthermore, in the realm of text-to-image generation, most papers use a combination of internal datasets. For example, Imagen (NeurIPS'22 Outstanding Paper) employs 440M internal data and 400M public data; ERNIE-ViLG (CVPR'23) utilizes LAION-5B and internal Chinese text-image pairs; MUSE (ICML'23) uses the same dataset as Imagen; and GigaGAN (CVPR'23) also leverages Adobe's internal data for its upsampler.
>
> **Q2: Model structure and hyperparameters. including VAE, and each stage of diffusion model.**
>
> The VAE model structure is based on the setup of Stable Diffusion, utilizing a KL-based VAE. Following the pipeline of LDM, an additional discriminator is introduced to train the VAE. And the downsampling ratio is 8, and z channel size is 12. We also change the ch (hyper-parameter of the LDM's VAE) from 128 to 256.
>
> Regarding the hyperparameters for the diffusion models, our approach consists of a single stage. We adhere to the UNet configurations of stable diffusion v2.1, with the exception of disabling the self-attention module in the largest resolution due to its computational complexity.
>
> We plan to include these details in our paper, and as soon as we gain access to edit it in openreview, we will upload the information accordingly.
>
> **Q3: Is the space-moe performed only on text-image cross attention? How about applying moe to self-attention within image?**
>
> The space-moe operation is specifically designed for text-to-image generation. Its purpose is to depict different *text* concepts within specific image regions. Thus, a *text* token is required to perform the cross-attention operation effectively.
>
> However, when it comes to the self-attention module, the self-attention map always resembles the "contour" of the image and does not have a direct mapping relationship to a particular token. As a result, the space-moe operation cannot be applied to the self-attention module. It can be applied to the cross-attention mechanism because it excels in capturing the correlations between textual descriptions and corresponding image regions.
>
> **Q4: In abstract the authors mentioned that "RAPHAEL exhibits superior performance in switching images across diverse styles" (L13). Is it benefit from spatial-moe? if so, is there any pattern in moe paths regarding different styles?**
>
> Yes, it benefits from space-moe.
>
> Firstly, as shown in Fig. 6b, space-moe significantly increases the CLIP score, which measures the alignment between images and text descriptions.
>
> Secondly, we observe some patterns (each concept generates 100 paths with our template). For style concepts such as anime/digital/realistic/cyberpunk/artistic/colorful/minimalist/bright, etc, we find that each pair in these styles (such as anime/digital, digital/realistic, etc) always share a relatively small number of experts (less than 7). Moreover, adjectives provided in our appendix also contain many style concepts, and they can be easily classified by XGBoost algorithm.
> So we think different style concepts have different diffusion paths.
>
> Thirdly, we conducted a human evaluation using the model with and without space-moe based on prompts containing style information from the ViLG-300 dataset. The results indicate that most people (76.15%) tend to prefer the model with space-moe, as they believe it better matches the images with the specified style.
>
> Finally, we also observe that diffusion paths reflect semantics, similar concepts/styles have similar diffusion paths. For example, smooth/glossy, minimal/minimalist, dreamy/dreamlike, happy/joyful, sad/gloomy, brave/courageous, tired/exhausted, bright/luminous, honest/sincere, puzzled/confused, brilliant/shining, grateful/thankful, harsh/severe, enormous/huge, humble/modest. We find the 200 diffusion paths generated by each pair share at least 11 experts out of 16 blocks. So we believe space-moe also helps the understanding of semantics.

---

> > ### Comment · Reviewer_8jJJ · 2023-08-20
> >
> > Thanks for your reply. All of my questions have been addressed, and I'm still leaning toward accepting this paper.

---

> > > ### Author Response · Authors · 2023-08-20
> > > **Thanks for your comments**
> > >
> > > We deeply thank you for the kind support of our work!

---

### Official Review · Reviewer_PszJ · 2023-07-07

**Soundness:** 4 excellent
**Presentation:** 3 good
**Contribution:** 3 good
**Rating:** 7
**Confidence:** 5

**Summary:**

This paper proposes RAPHAEL, a new text-to-image generative model, based on the latent diffusion model framework. The main methodological contribution is the use of space-mixture-of-experts (space-MoE) layers. These are layers that focus on different concepts from the text prompt in different spatial areas of the synthesized image. Different space-MoEs are automatically chosen for the different text tokens, and they are assigned to the relevant regions in the image, which can be found through the cross-attention maps. A similar time-MoE is also incorporated, although that is less novel. The model is validated on standard benchmarks and achieves state-of-the-art performance. Ablations over all relevant new components and hyperparameters are performed. The paper shows various qualitative results and model samples, which are quite impressive. The authors also demonstrate that RAPHAEL can be easily combined with ControlNet, LoRA fine-tuning, and super resolution GANs to enhance the image resolution.

**Strengths:**

The main strengths of the paper are:
- State-of-the-art text-to-image generation performance, including visually impressive results.
- Extensive ablation studies on all relevant components.
- The space-MoE idea is well motivated, novel, and boosts performance non-negligibly.
- I like in particular the gating mechanisms in both the space- and the time-MoE, which, for instance in the time-MoE case, automatically learn to assign different timesteps to various time experts. Previous work manually assigned experts to different diffusion time intervals (for instance, eDiff-I).

**Clarity:** The paper is well written and easy to read and follow. There are no major concerns regarding clarity. However, some details seem to be missing (see below).

**Originality:** In general, the mixture-of-experts idea is not new and has existed in language models and was also used in the text-to-image literature, for instance in the related eDiff-I (only time experts). However, the space-MoE technique specifically in the context of text-to-image generation is new, to the best of my knowledge, and it is well-motivated and seems to meaningfully boost performance. Hence, while the paper's originality is not groundbreaking, the main technical contribution is well-motivated and novel.

**Significance:** Text-to-image generation is a highly relevant and impactful topic, and RAPHAEL achieves state-of-the-art performance in this competitive area. Its mixture-of-experts approach is well-motivated and may find wider adoption. Apart from the quantitative evaluations, its visual results are stunning. Hence, I think the paper is impactful and significant.

**Quality:** The overall quality of the paper is high. The paper is easy to read, appropriately discusses the related literature, provides a background section, runs extensive ablation studies for all new relevant parameters, and supports its claims by appropriate experiments. The qualitative and quantitative results are strong. There are only relatively minor concerns with respect to missing details (see below).

**Weaknesses:**

The paper does not have any major weaknesses. However, I have some minor concerns:
1. Many details are missing:
    - The paper only mentions on the side (Line 192) that it is using a latent diffusion model framework and does not operate in pixel space.
       That is fine; however, this requires more details. For instance:
         - Was the autoencoder regularized? The LDM paper uses either KL-based or VQ-based regularization.
         - Was the autoencoder trained only with a reconstruction loss? Or also with a (patch-wise) discriminator?
         - What was the downsampling ratio?
   - The work should explain the multi-scale training in more detail. How exactly is the model trained at all these different resolutions and aspect ratios at once?
   - The gating mechanisms incorporate an $argmax$ function. The $argmax$ is usually not differentiable. How did the authors deal with that, to enable regular backpropagation for training? I think it would be helpful to discuss this in a bit more detail.
   - The paper says that RAPHAEL is trained on LAION "and a few internal data". What is this internal data? Even if this data is internal and not released, the authors should describe it and what value it brings on top of LAION. E.g. what is the size of that internal data? Does it have any special characteristics? Only high-quality, for instance? Etc.
2. A discussion on limitations is missing. This would further strengthen the paper.

In conclusion, the paper's main weaknesses are all related to missing details and I believe these issues can be addressed easily. I do not see any other major concerns. Considering the paper's strengths discussed above, I am consequently suggesting acceptance of the paper.

**Questions:**

I have only one minor question (just curiosity, not impacting the paper rating):

Figure 4 shows the diffusion paths for different simple concepts. Do more related or similar concepts also have more similar diffusion paths? For instance, would "strawberry" and "raspberry" share much of their paths, while "strawberry" and "car" would not?

**Limitations:**

Potential negative societal impacts have been briefly, but sufficiently addressed. Limitations have not been discussed. What are RAPHAEL's limitations? I would like to encourage the authors to add a critical discussion on this.

---

> ### Author Rebuttal · Authors · 2023-08-08
>
> Dear Reviewer PszJ,
>
> Thank you for appreciating our approach. We will address your concerns below.
>
> **Q1: Details of VAE**
>
> Yes, we follow the setup of Stable Diffusion and use KL-based VAE. When training the VAE, we add an extra discriminator following the pipeline of LDM. And the downsampling ratio is 8, z channel size is 12. We also change the ch (hyper-parameter of the LDM's VAE) from 128 to 256.
>
> **Q2: The work should explain the multi-scale training in more detail. How exactly is the model trained at all these different resolutions and aspect ratios at once?**
>
> Thanks for your question. I will explain it in detail:
> As outlined in our research paper, we employ a system comprising 9 buckets, each representing a distinct image scale. The initial step involves resizing an image to its nearest size within these predefined buckets. Subsequently, the allocation of GPU resources to each bucket is automated, based on the number of images they contain. This approach ensures efficient utilization of computational resources. Here is a step-by-step breakdown of the process:
>
> 1. Aspect Ratio List ($R$): We maintain a list ($R$) containing aspect ratios for all the images in our dataset. The length of this list corresponds to the total number of images.
> 2. Bucket List ($L$): We establish nine buckets ($L$) that encompass various image sizes, including  [448, 832], [512, 768], [512, 704], [640, 640], [576, 640], [640, 576], [704, 512], [768, 512], and [832, 448]. For a given image ratio in $R$, we identify the nearest bucket size in $L$ through a matching process. For example, images with an aspect ratio of 1.0 will be associated with the bucket of size [640, 640].
> 3. Mapping Aspect Ratios to Buckets ($R1$): As a result of the previous step, we generate another list ($R1$) of the same size as $R$. Each element in $R1$ indicates the bucket to which the corresponding element from $R$ is assigned.
> 4. GPU Allocation: We proceed to compute the GPU allocation based on the information provided by $R1$. Firstly, we calculate the total number of images each bucket contains. Using this information, we create another list ($L1$) of the same size as $L$, and L1.sum() denotes the total images we have. Then, we utilize a simple trick to distribute these buckets across different GPUs. The following code snippet demonstrates this process: We provide codes below:
>
>    bk_gpu_nums = np.clip((L1 / L1.sum() * world_size).astype(int), 0, world_size),
>
>    bk_gpu_nums[bk_gpu_nums.argmax()] = bk_gpu_nums[bk_gpu_nums.argmax()] - (bk_gpu_nums.sum() -world_size),
>
>    Here, "world_size" denotes the total number of GPUs available, and "bk_gpu_nums" signifies the number of GPUs required for each bucket. Then we assign different GPUs to different buckets.
> 5. Image Selection: Each GPU within a specific bucket will select images from the dataset according to the mapping provided by $R1$, and this GPU will train the model with this bucket scale.
>
> While we acknowledge that the "astype(int)" operation may involve approximation, it becomes negligible when dealing with a large number of GPUs and datasets.
>
> Finally, it is essential to note that each GPU employs the same batch size during the training process.
>
> **Q3: Implementation of $argmax$ function**
>
> Given a vector $v$, the first step is to compute the softmax value $y_{soft} = softmax(v)$.
>
> Next, the gating functions are determined using the $argmax$ operation, resulting in output $y_{hard} = argmax \ y_{soft}$.
>
> To ensure differentiability and enable backpropagation during training, we adopt a technique that bridges the gap between the discrete and differentiable representations. We introduced a soft version of the gating function, denoted as $y_{hard'}$, which can be expressed as follows: $y_{hard'} = (y_{hard} - y_{soft}).detach() + y_{soft}$.
>
> By using this softened version of the gating function, we could successfully perform backpropagation during training. Although the gating decisions were made based on $y_{hard'}$, the actual backpropagation process was executed on the differentiable representation $y_{soft}$.
>
> **Q4: Internal data**
>
> We have $\approx$ 440M data filtered from LAION-5B, and $\approx$ 290M internal datasets, which is less than Imagen's. Yes, it has special characteristics, such as high-quality, high-aesthetics.
>
> To collect our internal datasets, we follow the methodology of DALL-E [1]. We curate a dataset on a scale similar to JFT-300M by sourcing images from the Internet. We remove instances with aspect ratios outside the range of [1/2,2], and we follow Stable Diffusion v1.4 to filter out images with low aesthetics scores. For captioning these images, we utilized BLIP-2. The main reason for constructing such an internal dataset with high aesthetics is to compensate for the poor quality of LAION.
>
> Furthermore, most papers on text-to-image generation use a series of internal datasets. For example, Imagen (NeurIPS'22 Outstanding Paper) uses 440M internal data and 400M public data; ERNIE-ViLG 2.0 (CVPR'23) uses LAION-5B and internal Chinese text-image pairs; MUSE (ICML'23) uses the same dataset with Imagen; GigaGAN (CVPR'23) also uses Adobe's internal data for its upsampler.
>
> **Q5: More properties of diffusion paths**
>
> Yes, we observe that similar concepts have similar diffusion paths (each concept generates 100 paths with our template). For example, smooth/glossy, minimal/minimalist, dreamy/dreamlike, happy/joyful, sad/gloomy, brave/courageous, tired/exhausted, bright/luminous, honest/sincere, puzzled/confused, brilliant/shining, grateful/thankful, harsh/severe, enormous/huge, humble/modest, etc. We find the 200 diffusion paths generated by the similar concept pair always share at least 11 experts out of 16 blocks.
>
> But for different concepts, such as strawberry/car, they are **different** things always share a small number of experts (less than 7).
>
> **Q6: Limitations**
>
> Please refer to the Q9 of reviewer NLKN, and we have discussed limitations in detail.

---

> > ### Comment · Reviewer_PszJ · 2023-08-18
> > **Thank you for the rebuttal**
> >
> > I would like to thank the authors for their rebuttal and for providing extensive details to answer my questions. It would be great to incorporate all these explanations into the final version of the paper. I do not have any further questions or concerns. I have been already positive about the paper and still suggest acceptance.

---

> > > ### Author Response · Authors · 2023-08-19
> > > **Thanks for your comments**
> > >
> > > We sincerely thank the reviewer for the kind support of our work! We will incorporate the details into our final version.

---

### Author Rebuttal · Authors · 2023-08-08

General response:
We express our sincere appreciation to all the reviewers for their valuable time and efforts in reviewing our paper. We are delighted to learn that the reviewers have generally recognized and appreciated the contributions made in our work, which include:

1. State-of-the-art performance (PszJ, 8jJJ, NLKN, and fnVX).
2. Introducing two novel ideas, namely, space-moe (PszJ, NLKN) and edge-supervised learning (NLKN).
3. Extensive ablation study (PszJ, 8jJJ, fnVX) and visualization results (8jJJ, NLKN).
4. Thorough comparison to other methods (NLKN).

We extend our gratitude to all the reviewers for their insightful and constructive suggestions. The main concerns raised by the reviewers revolve around our experimental settings. To address these concerns, we provide the following summaries and discuss them in rebuttal:
1. Internal datasets (PszJ, 8jJJ, NLKN, fnVX).
2. Multi-scale training implementation (PszJ, NLKN).
3. VAE and U-Net configurations (PszJ, 8jJJ, NLKN).
4. Ablation study settings (NLKN).
5. Choice of Focal Loss (NLKN).
6. Additional properties of diffusion paths. (PszJ, 8jJJ, NLKN)
7. Inference cost. (fnVX)

We are committed to incorporating these improvements and addressing all the raised concerns in our revised paper once we can edit it in openreview.

We will also make a global response for the Q1,Q2,Q3 above:

**Q1: Information about internal dataset**

The data consists of approximately 440 million entries filtered from LAION-5B and approximately 290 million entries from internal datasets, which is less than what Imagen's dataset contains. These datasets possess special characteristics, notably high-quality and aesthetics.

To collect our internal datasets, we follow the methodology of DALL-E. We curate a dataset on a scale similar to JFT-300M by sourcing images from the Internet. We remove instances with aspect ratios outside the range of [1/2,2], and we follow Stable Diffusion v1.4 to filter out images with low aesthetics scores. For captioning these images, we utilized BLIP-2. The main reason for constructing such an internal dataset with **high aesthetics** is to compensate for the poor quality of LAION. We don't add additional limits on categories but want to collect datasets with high aesthetics.

The resolution distribution is [52,24,161,470,52,81,56,70,34] if we adopt the buckets in Q2 and 1k GPUs. It indicates, for example, there are 52 GPUs for [448,832] and each GPU shares a similar number of images.

**Q2: Implementation of multi-scale training**

As outlined in our research paper, we employ a system comprising 9 buckets, each representing a distinct image scale. The initial step involves resizing an image to its nearest size within these predefined buckets. Subsequently, the allocation of GPU resources to each bucket is automated, based on the number of images they contain. Here is a step-by-step breakdown of the process:

1. Aspect Ratio List ($R$): We maintain a list ($R$) containing aspect ratios for all the images in our dataset. The length of this list corresponds to the total number of images.
2. Bucket List ($L$): We establish nine buckets ($L$) that encompass various image sizes, including **[448, 832], [512, 768], [512, 704], [640, 640], [576, 640], [640, 576], [704, 512], [768, 512], and [832, 448]**. For a given image ratio in $R$, we identify the nearest bucket size in $L$ through a matching process. For example, images with an aspect ratio of 1.0 will be associated with the bucket of size [640, 640].
3. Mapping Aspect Ratios to Buckets ($R1$): As a result of the previous step, we generate another list ($R1$) of the same size as $R$. Each element in $R1$ indicates the bucket to which the corresponding element from $R$ is assigned.
4. GPU Allocation: We proceed to compute the GPU allocation based on the information provided by $R1$. Firstly, we calculate the total number of images each bucket contains. Using this information, we create another list ($L1$) of the same size as $L$, and L1.sum() denotes the total images we have. Then, we utilize a simple trick to distribute these buckets across different GPUs. The following code snippet demonstrates this process: We provide codes below:

   bk_gpu_nums = np.clip((L1 / L1.sum() * world_size).astype(int), 0, world_size),

   bk_gpu_nums[bk_gpu_nums.argmax()] = bk_gpu_nums[bk_gpu_nums.argmax()] - (bk_gpu_nums.sum() -world_size),

   Here, "world_size" denotes the total number of GPUs available, and "bk_gpu_nums" signifies the number of GPUs required for each bucket. Then we assign different GPUs to different buckets.
5. Image Selection: Each GPU within a specific bucket will select images from the dataset according to the mapping provided by $R1$, and this GPU will train the model with this bucket scale.

While we acknowledge that the "astype(int)" operation may involve approximation, it becomes negligible when dealing with a large number of GPUs and datasets.

Finally, it is essential to note that each GPU employs the same batch size during the training process. This comprehensive approach enables effective multi-scale training, resulting in enhanced performance and robustness of the model.

Given 1000 GPUs and the predefined buckets, the final resolution distribution (including LAION and internal datasets) of these buckets is [28,67,116,419,43,52,76,112,87], which means there are 28 GPUs in bucket [448, 832].

Hence, the different bucket has different images and different number of GPUs. Each GPU will sample from its belonging bucket. The VAE and diffusion models both need multi-scale training.

**Q3: Configs of VAE and U-Net**

For U-Net, the difference between ours and Stable Diffusion v2.1 is we disable the self-attention module for the largest resolution due to its computation complexity. For VAE, the downsampling ratio is 8, z channel size is 12. We also change the ch (hyper-parameter of the LDM's VAE) from 128 to 256.

---

### Decision · Program_Chairs · 2023-09-21

**Decision:**

Accept (poster)

**Comment:**

This paper received all acceptance scores, with two acceptances, one weakly acceptance, and one borderline acceptance. The paper introduces RAPHAEL, a new text-to-image generative model, based on the latent diffusion model framework, and it is validated on standard benchmarks and achieves state-of-the-art performance. Moreover, RAPHAEL is a technically solid work with its valuable contributions to the community. As a result, AC has decided to accept this paper.